# ROS homeostasis mediated by MPK4 and SUMM2 determines synergid cell death

Ronny Völz [1✉], William Harris [2], Heribert Hirt [3,4] & Yong-Hwan Lee [1,2,5,6,7✉]

Sexual plant reproduction depends on the attraction of sperm-cell delivering pollen tubes (PT) by two synergids, followed by their programmed cell death (PCD) in *Arabidopsis*. Disruption of the mitogen-activated protein kinase 4 (MPK4) by pathogenic effectors activates the resistance protein (R) SUMM2-mediated immunity and cell death. Here we show that synergid preservation and reactive oxygen species (ROS) homeostasis are intimately linked and maintained by MPK4. In *mpk4*, ROS levels are increased and synergids prematurely undergo PCD before PT-reception. However, ROS scavengers and the disruption of SUMM2, in *mpk4*, restore ROS homeostasis, synergid maintenance and PT perception, demonstrating that the guardian of MPK4, SUMM2, triggers synergid-PCD. In *mpk4/summ2*, PTs show a *feronia*-like overgrowth phenotype. Our results show that immunity-associated PCD and synergid cell death during plant reproduction are regulated by MPK4 underscoring an underlying molecular mechanism for the suppression of plant reproduction during systemic R-mediated immunity.

[1] Research Institute of Agriculture and Life Sciences, Seoul National University, Seoul 08826, Korea. [2] Department of Agricultural Biotechnology, Seoul National University, Seoul 08826, Korea. [3] Center for Desert Agriculture, Division of Biological and Environmental Sciences and Engineering, King Abdullah University of Science and Technology, Thuwal 23955-6900, Saudi Arabia. [4] Max F. Perutz Laboratories, University of Vienna, 1030 Vienna, Austria. [5] Center for Fungal Genetic Resources, Seoul National University, Seoul 08826, Korea. [6] Plant Immunity Research Center, Seoul National University, Seoul 08826, Korea. [7] Center for Plant Microbiome Research, Seoul National University, Seoul 08826, Korea. ✉email: Ronnynabu@snu.ac.kr; yonglee@snu.ac.kr

In angiosperms, germ cells are formed during the haploid stage of a plant's life, in structures termed female gametophytes (FG)[1], which are embedded in diploid, sporophytic tissue. Within the FG, eight nuclei differentiate to the gametic egg and central cell (CC) adjoined by two kinds of accessory cell types, called synergids and antipodal cells. Both, the haploid egg and adjacent homodiploid CC are each fertilized by one sperm cell to form the embryo and the surrounding triploid endosperm, respectively, within the polygonum gametophyte. Even though the gametophyte carries a haploid set of chromosomes, the CC bears a bi-nuclear ploidy that originates from the karyogamy of two polar nuclei. Besides the two gametic cells, two synergids lie at the micropyle and coordinate the attraction of the pollen tube (PT) by the discharge of attractants[2]. The synergids are subdivided into the receptive and non-receptive/persistent synergid. The receptive synergid undergoes programmed cell death (PCD) instantly during PT reception and PT rupture, whereas the non-receptive synergid persists[3].

Sexual reproduction of flowering plants is based upon the tight coordination of the fertilization process including the synergid-mediated PT attraction[4,5], degeneration of the receptive synergid, and the elimination of the persistent synergid via the cell fusing with the emerging endosperm after egg and CC fertilization[6].

Mitogen-activated protein (MAP) kinase cascades exert critical roles in plant immunity[7,8]. A distinct feature of MAPK-mediated signaling is that three layers of protein kinases form a sequential phosphorylation cascade that consists of a MAP kinase kinase kinase (MAPKKK), a MAP kinase kinase (MKK), and a MAP kinase (MPK). MPK4 is a multifunctional kinase that regulates various signaling events in plant innate immunity, defense response, and cytokinesis[9]. Initially, MPK4 was identified as a negative regulator of plant immunity[10]. The *mpk4* mutant shows a strong autoimmune phenotype accompanied by dwarfism, hydrogen peroxide accumulation, spontaneous cell death, and a deviating transcriptomic and alternative splicing profile[11,12]. The MPK4-cascade is induced upon the perception of pathogen-associated molecular pattern (PAMP) recognized by PAMP receptors upstream of MEKK1. The upstream signaling components MEKK1 and MKK1/MKK2 activate MPK4 by phosphorylation[13] thereby coordinating basal resistance and effector-triggered immunity (ETI)[14]. The integrity of the MPK4 cascade is guarded by the R protein nucleotide-binding leucine-rich repeat (NB-LRR) protein SUPPRESSOR OF *mkk1 mkk2* (SUMM2). SUMM2 monitors the phosphorylation of MPK4 substrates such as CALMODULIN-BINDING RECEPTOR LIKE KINASE 3 (CRCK3/SUMM3)[15]. CRCK3 associates with SUMM2 in *planta* and might act as a decoy or guardian of SUMM2. Inactivation of MPK4 by pathogenic effectors, e.g. the HopAI1 of *Pseudomonas*, reduces CRCK3 phosphorylation, and eventually activates SUMM2[16]. Activated SUMM2 triggers autoimmunity and PCD, as observable in the MPK4-deficient mutant. MPK4 plays a role in plant cytokinesis and is required for the formation of the cell-plate[17]. Intriguingly, homozygote *mpk4* plants exhibit a severe defect in male-specific meiotic cytokinesis[18]. The microspore mother cells fail to form a normal intersporal callose wall after male meiosis, and cannot complete the meiotic cytokinesis.

Here, we present that MPK4 function is vital for synergid preservation and PT reception via the suppression of the immune receptor SUMM2-mediated ROS production and cell death reaction.

## Results

**Premature synergid cell death in *mpk4*.** The immune-activated MAP kinase 4 (MPK4) functions as a suppressor of PCD in plants and its homozygous disruption is accompanied by a strong autoimmune reaction and plant sterility. Notably, PCD is a critical element during fertilization[19]. To unravel the role of the MPK4-kinase cascade in the process of plant reproduction, we analyzed the *mpk4-2*[20] and *mpk4-3* mutants[13] through the course of female gametogenesis and seed formation compared to wild type (WT) and a *pMPK4::MPK4:GFP* complementation line[20] in the *mpk4-2* background.

To determine the PT attraction ability of *mpk4*, we pollinated WT, *mpk4*, and the complementation line with a PT marker carrying *LAT52::GUS* originating from a WT source[21]. After successful PT attraction and reception by the synergids, the blue GUS staining is observable in the respective FGs. We detected GUS-positive ovules in approximately 97-98% of WT and the complementation line samples (Fig. 1a). However, in *mpk4-2* only 3.9% and in *mpk4-3* 4.12% of FGs showed a blue GUS staining. This outcome suggests a defect of *mpk4* in PT attraction. Yet, we could not rule out that the PTs are affected on their journey through the transmitting tract and, thus, cannot arrive at the FG. Therefore, we analyzed the ability of WT PTs to grow through the transmitting tract in WT and *mpk4* by carrying out aniline blue staining and GUS-stained PT-tracking. We found that WT PTs are fully capable of growing within the transmitting tract in WT and *mpk4* and consequently reaching the base of the ovule, the funiculus, in a comparable manner with minor differences (Fig. 1b, c and Supplementary Fig. 1m, n). However, the PTs in *mpk4*, after having approached the funiculus, did not grow further towards the ovules and seemed to lose guidance (Fig. 1c, e, f). This result suggests a defect in the short-track PT attraction in *mpk4*. To evaluate this finding, we performed a semi-in-vitro PT attraction assay. We applied WT pollen on a WT stigma and tracked the PT attraction ability of WT and *mpk4* ovules which were placed in close vicinity to the stigma. In about 74% of cases the PTs were successfully attracted by the WT ovules, in contrast to *mpk4* ovules, which predominantly failed to attract the PTs, with a perception rate of only 3.5% (Fig. 1d, g, h). Together, these findings show that the PT growth towards the ovules is not compromised by the *mpk4* mutation, but suggests that the short-track PT-attraction might be affected which is mediated by the synergids of the FG.

Thus, we raised the question of whether the formation of the female gametophytic cells is compromised in *mpk4*. Before PT-reception and fertilization, we found in *mpk4* that the egg cell, CC, and antipodal cells are morphologically indistinguishable from WT and the complementation line (Fig. 1i, j, o, p, r, Supplementary Fig. 1h, j and Supplementary movie 1). The CC showed a slight increase of unfused polar nuclei in *mpk4* (Fig. 1n, r and Supplementary Fig. 1h, i). However, in the vast majority of *mpk4* FGs, we detected a severe defect in the maintenance of the synergid nuclei. In the maximum projection of confocal-laser scanning microscopy (CLSM) Z-stack observations, we detected bright white areas in the synergids which verifies synergid degeneration[22,23] (Fig. 1k-m, R and Supplementary movie 2, 3). Moreover, synergids could barely be recognized at all after sample stack observation, whereas the detection of the egg and CC was comparable to WT (Fig. 1n–p, r, Supplementary Fig. 1h-j and Supplementary movie 4). In summary, we predominately observed either only a single synergid nucleus or no synergid nuclei at all, 48 h post-emasculation.

To analyze the synergid degeneration at the molecular level, we introduced a female-gametophyte reporter (FGR7.0) to *mpk4*. This marker allows simultaneous characterization of synergid cell, egg cell, and CC in a single construct[23]. Additionally, we analyzed an antipodal marker in *mpk4* to characterize these accessory cells. We found the egg cell (*pEC1::NLS:dsRED*), CC (*pDD22::YFP*), and antipodal (*pAGL62::AGL62:GFP*) marker expressions indistinguishable from WT (Fig. 2a–c), Supplementary Fig. 1e, f)

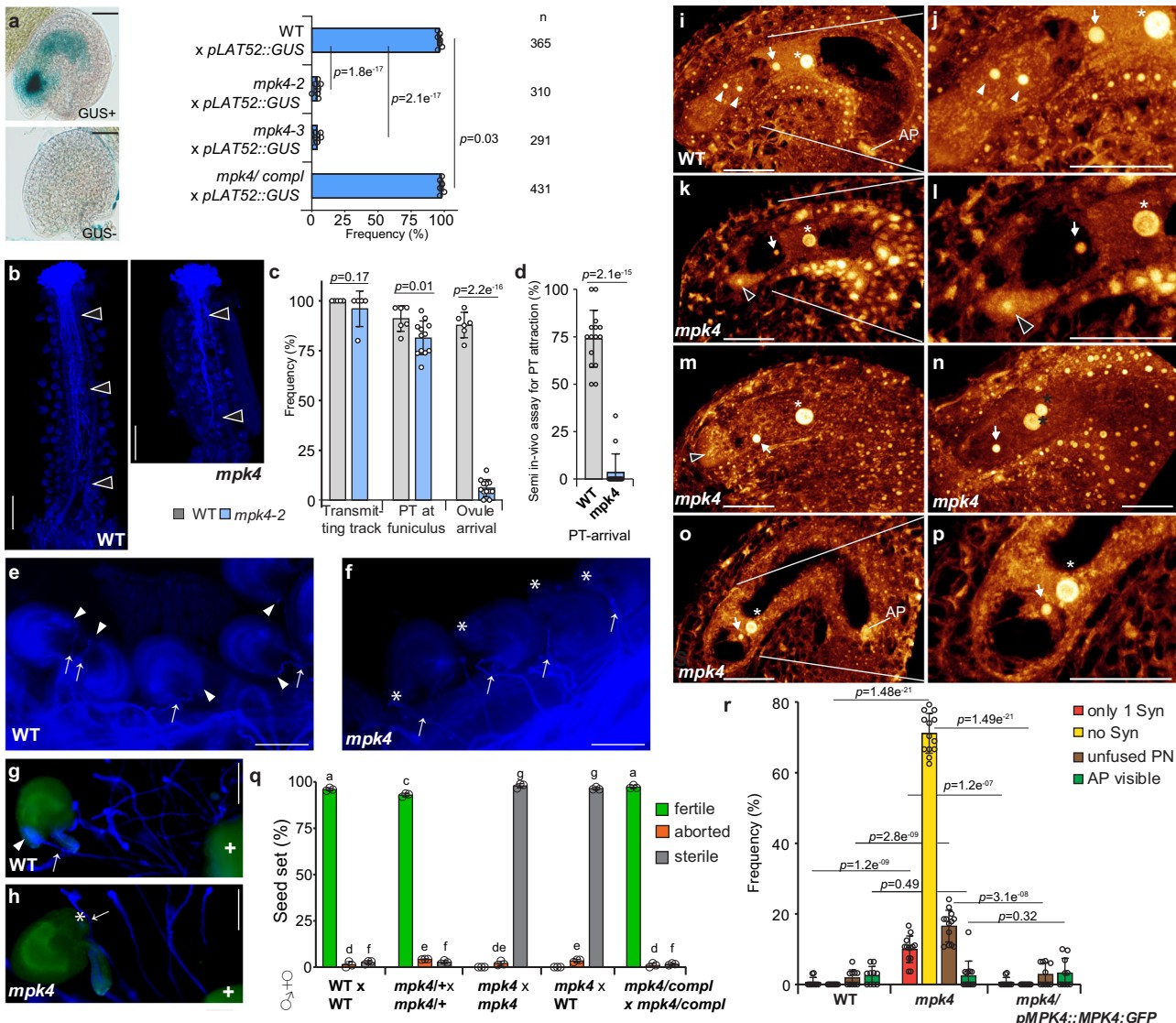

**Fig. 1 Premature synergid degeneration in *mpk4*. a** Pollen tube (PT) attraction assay in wild-type (WT), *mpk4-2*, *mpk4-3*, and the complementation line expressing *pMPK4::MPK4GFP* in *mpk4-2*. Plants were pollinated with pollen of *pLAT52::GUS*-expressing plants. PT attraction was analyzed 24 hr after pollination. Scale bars, 30 μm. **b–c, e–f** PT growth assay through the transmitting tract to the ovules in WT and *mpk4-2* by aniline blue staining (**b**, scale bar 0.5 mm; **e**, **f**, scale bar 0.1 mm). WT pollen was applied and the growth through stigma, transmitting track towards the ovule was monitored for 24 h. Presented are the results of the 5 biological replicates (*n* = 26). white-edged black arrowhead, PTs; white arrowhead, PT arrived at the FG; white asterisk, unguided PT; white arrow, PT on the funiculus/ close to the ovule. Error bars show ± SD. (**d, g, h**) Analysis of the short-track PT attraction ability in WT and *mpk4* by semi-in vivo PT assay followed by PT calcofluor white staining. WT pollen were applied on WT-stigmas and ovules of WT (**g**) and *mpk4* (**h**) were placed below the stigma decapitation site. The PT attraction ability was analyzed after 6 h. White plus refers to the decapitated WT stigma; white arrowhead, PT arrived at the FG; white asterisk, unguided PT; white arrow, PT on the funiculus/ close to the ovule. Presented are the results of 3 biological replicates comprising a total of 69 ovules in WT and 68 ovules in *mpk4*. Error bars show ± SD. Scale bar 0.1 mm. **i–p** Representative CLSM images of WT (**i–j**) and *mpk4* (**k–p**) female gametophytes (FG) 48 h post-emasculation. Depicted are maximum projections of Z-stacks, with a focal plane distance of 1 μm within a total range of about 20 μm, animated 3D-projections are provided in Supplementary movies 1-4. **k–m** FG of *mpk4* that shows prematurely degenerating synergids. **n–p** FG of *mpk4* with completely absent synergids. **n** FG with polar nuclei and without synergids. White arrowhead, synergid; black arrowhead, degenerating synergid; white arrow, egg cell; white asterisk, central cell nucleus; black asterisks, polar nuclei. Scale bars, 30 μm. **q** Seed formation in WT (*n* = 287), *mpk4/+* (*n* = 516), *mpk4/-* (*n* = 331), *mpk4/- x* WT (*n* = 225) and the MPK4 complementation line (*n* = 249). Error bars show ± SD. Statistical significance was analyzed by one-way ANOVA. Letters indicate significant differences from the mean (*p* ≤ 0.05) within the same category of either fertile, aborted or sterile among the different genotypes. **r** Morphological characterization of the female gametophytic cells in WT (*n* = 314), *mpk4-2* (*n* = 366) and *mpk4* complementation line (*n* = 306) 48 h post-emasculation. Error bars show ± SD. Syn, Synergids; PN, polar nuclei; AP, antipodal cells. **a, c, d, r** Statistical significance was analyzed by one-way ANOVA. Source data and further statistical analysis are provided in the source data file.

which suggests that the development and the molecular profile of these gametophytic cells are not broadly perturbed. However, the synergid marker, driven by the regulatory sequence of *AT5G43510/DD2*[24,25], was barely detectable and could be observed in only a small minority of FGs (Fig. 2a–c). Strikingly,

the majority of FGs completely lacked the synergid marker expression. Likewise, we analyzed the expression of *pHD2B::GFP:HD2B*[26] which was predominantly found in synergids and CC in WT. However, in *mpk4*, GFP:HD2B could hardly be captured in the synergids (Supplementary Fig. 1a, b, d).

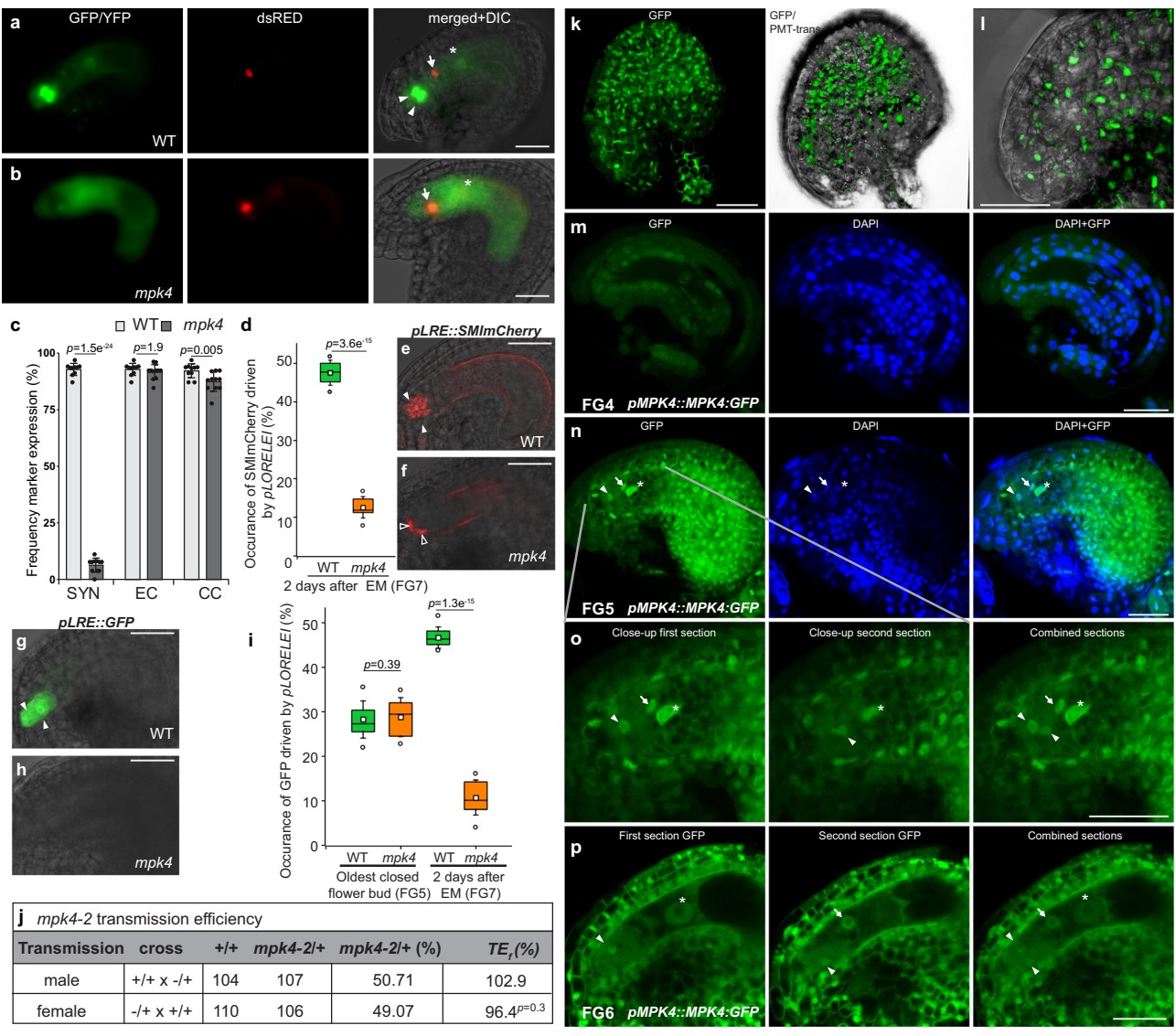

**Fig. 2 Premature synergid degeneration at the molecular level, and MPK4 localization in the embryo sac. a–c** Expression of the female-gametophyte reporter (FGR7.0) in (**a**) WT and (**b**) *mpk4*. c Frequency of *FGR7.0/+* expression in WT ($n = 321$) and *mpk4* ($n = 285$) ovules was determined. Error bars show ± SD. **d–i** Expression of the *LORELEI*-reporter (*pLRE*) coupled to the Golgi-retention peptide sequence SMImCherry (**d–f**) and a single GFP (**g–i**). The expression of *pLRE::GFP/-* was analyzed in the oldest-closed flower bud (FG5 stage) (WT, $n = 326$; *mpk4*, $n = 240$) and 48 h after emasculation (WT, $n = 328$; *mpk4*, $n = 251$; FG7 stage). *pLRE::SMImCherry/-* was analyzed 48 h after emasculation (WT, $n = 293$; *mpk4*, $n = 257$). Boxes represent the 25th and 75th percentiles, and the inner rectangle highlights the median, whiskers show the SD, and outliers are depicted by dots (Min/max range). **j** Transmission analysis of *mpk4*. Transmission efficiency (TE$_f$) indicates the transmission of *mpk4* through the male and female gametophyte.
**k–p** Representative images of MPK4:GFP expression and localization in the sporophytic ovule tissue (**k–l**) and at the female gametophytic developmental stages FG4, FG5, FG6 (**m–p**). **k–l** Depicted are maximum projections of Z-stacks, with a focal plane distance of 2 μm within a total range of about 30 μm, animated 3D-projection is provided in Supplementary movie 5. **m–p** *pMPK4::MPK4:GFP*-expressing line was analyzed, accompanied by the nuclei-staining dye DAPI. Negative control Supplementary Fig. 1l. Experiments were repeated three times with similar results and presented are representative images.
**c**, **d**, **i**, **j** Statistical significance was analyzed by one-way ANOVA. Source data and further statistical analysis are provided in the source data file. White arrowhead, synergid; black arrowhead, degenerating synergid; white arrow, egg cell; white asterisk, central cell nucleus; black asterisks, polar nuclei. Scale bars, 20 μm (**a**, **b**, **e**, **f–h**), 30 μm (**k–p**).

Moreover, we analyzed the expression of the synergid-expressed gene *LORELEI* with a *pLRE::GFP* (Fig. 2g–i) and *pLRE::SM1-mCherryGolgie* reporter (Fig. 2d–f)[27]. Both reporters were only detected in a minority (10–12%) of *mpk4* synergids compared to WT. To analyze whether the synergids initiation in *mpk4* resembles WT, we analyzed the *pLRE::GFP/-* reporter at synergid establishment in the oldest closed flower buds (Fig. 2i) corresponding to the FG5 stage in the FG development[22]. In WT and *mpk4*, we captured a GFP signal in about 30% of synergids, indicating a comparable synergid formation in WT and *mpk4*.

Our results demonstrate that before PT reception and fertilization have taken place, synergids in *mpk4* are established per se but prematurely degenerate. We conclude that MPK4 guards synergids against premature PCD and thereby enables PT attraction, sperm-cell release, and gamete fertilization.

Thus, we raised the question of whether the premature synergid degeneration in *mpk4* interferes with seed formation. In *mpk4* self-pollinated plants, fertile seed formation could not be observed, solely an amount of 2.11% aborted seeds ($n = 331$) could be counted (Fig. 1q). This finding is following previous

results showing the inability of homozygote *mpk4* plants to reproduce. To analyze whether WT pollen might increase the fertilization rate, we hand-pollinated *mpk4* with WT pollen. Again, we could not detect fertile seeds and only 3.56% of embryo sacs initiated seed formation before aborting, these were accompanied by an overwhelming number of sterile embryo sacs. As a matter of note, in the few *mpk4* gametophytes that truly received a PT, gamete fertilization, embryo, and endosperm initiation were not broadly perturbed suggesting that egg and CC maturation are not adversely affected by the *mpk4* knock-out (Supplementary Fig. 3g). The introduction of a complementation construct (*pMPK4::MPK4:GFP*)[20] in *mpk4* restored seed formation indicating that the lack of seed production can be traced back to the knock-out of MPK4. Our results indicate a vital function of MPK4 in guarding the synergids against premature degeneration thereby enabling PT attraction and seed formation.

**Transmission analysis**. To determine whether the premature synergid degeneration is of sporophytic or gametophytic origin, we analyzed the heterozygote *mpk4/+* mutant. The occurrence of the synergids, central cell and antipodals was indistinguishable from WT (Supplementary Fig. 3a–c and Supplementary movie 7, 8). Likewise, the expression of the *pLRE::SM1-mCherryGolgie* reporter in the synergids did not show a significant deviation from WT (Supplementary Fig. 3d, e). The transmission efficiency of self-crossed heterozygote *mpk4* mutants resulted in a Mendelian segregation ratio of 28.4% WT, 53.5% heterozygote and 18.1% homozygote *mpk4* mutants, suggesting a weak zygotic/ embryonic lethal phenotype (Supplementary Fig. 3f). Indeed, when compared to WT, the seed set of heterozygote *mpk4* plants showed a slight reduction of fertile seeds in favor of a slight increase of aborted seed formation (Fig. 1q). To further analyze, whether the transmission of the *mpk4* mutation through the male and female gametophyte is affected, we performed reciprocal crosses between WT and heterozygote *mpk4* mutants and analyzed the *mpk4* transmission among the F1 progeny. Both, when *mpk4* was introduced from either the male or female site, the transmission of the *mpk4* mutation to the next generation was determined very close to 50%, equating a transmission efficiency of almost 100% (Fig. 2j). This result indicates that the gametophytic *mpk4* transmission is not affected. Taken together, our results point to a sporophytic origin of the premature synergid degeneration in *mpk4/-* revealing that the surrounding maternal ovule tissue might specifically determine the formation and function of the PT-attracting synergids.

**MPK4:GFP localization in the female gametophytic cells and ovule tissue**. To pinpoint the expression and localization of MPK4 in the ovular tissue and during female gametogenesis, we analyzed the line expressing *pMPK4::MPK4:GFP*[20] by CLSM and exploited maximum projections of Z-stack observations. MPK4:GFP showed a strong nuclear signal in the FG-surrounding sporophytic ovule tissue, including the micropylar region of the ovule (Fig. 2k, l and Supplementary movie 5). During the early female gametogenesis at stage FG4, we could not detect MPK4:GFP in the syncytial nuclei (Fig. 2m). However, after cellularization at stage FG5 and FG6, MPK4:GFP was detected in the nucleus of the synergids, egg cell and CC and co-localized with the nucleus-staining agent DAPI (Fig. 2n–p and Supplementary Fig. 1k). In addition, we found MPK4 localization not only in the nucleus but also in the cytoplasm of the female gametophytic cells, particularly in the synergids (Fig. 2p). The negative control by the use of WT plants, did not exhibit a signal under the chosen parameters for CLSM (Supplementary Fig. 1l).

These results revealed the expression of *MPK4* and the localization of the encoded protein in the cellularized female gametophyte and the surrounding ovular sporophytic tissue.

**Reactive oxygen species accumulation in *mpk4* synergids**. Hydrogen peroxide ($H_2O_2$) and superoxide are the main triggers for the immune- and developmental-regulated PCD in plants[28–32]. Previous studies have revealed that synergid degeneration occurs as a PCD response[5,23] and ROS is well known to trigger PCD in plant cells[33]. However, whether ROS triggers synergid PCD still needs to be deciphered. Furthermore, the MPK4-cascade is essential for ROS-homeostasis and signaling[9,34]. These previous findings prompted us, to analyze whether the premature synergid cell death in *mpk4* might be triggered by a deviated ROS homeostasis. Nitroblue tetrazolium (NBT) is the most commonly employed probe for the detection of in situ superoxide. The NBT staining in WT showed a moderate superoxide accumulation in the inner sporophytic integument cells of the ovule and throughout the female gametophyte within the synergid, egg cell, and CC (Fig. 3a–c). However, in *mpk4*, we detected very intense NBT staining in the micropylar ovule region while the NBT accumulation in the remaining gametophytic cells was less frequent compared to WT. The sporophytic superoxide concentration in *mpk4* was reminiscent of the results in WT (Fig. 3c). To detect in-situ $H_2O_2$ levels, we performed diamino-benzidine (DAB) staining. The insoluble brown precipitate that refers to cellular levels of $H_2O_2$ could be detected in *mpk4* in the egg cell and CC region and the sporophytic ovule tissue in a comparable manner to WT (Fig. 3d–f). Yet, intensely dark DAB staining was mainly captured in *mpk4* in the filiform apparatus and the nearby micropylar ovule tissue. Additionally, we applied $H_2$DCF-DA to evaluate the amount of hydroxyl radicals. We detected moderate fluorescence signals in WT synergids, prevalently at the filiform apparatus (Fig. 3g, h). However, the signal intensity was massively enhanced at the micropylar region in *mpk4* (Fig. 3i, j, o). These results indicate sustainably higher amounts of the cell-death agents $H_2O_2$, superoxide radicals and hydroxyl radicals in the micropylar embryo sac/ synergid cell region in *mpk4* that might drive synergid PCD thereby indicating MPK4 as repressor of ROS metabolism in the micropylar ovule region.

To decipher whether indeed enhanced ROS levels in *mpk4* trigger premature synergid PCD, we took advantage of a pistil-feeding approach to manipulate the synergid/ovule environment[35]. We firstly applied diphenyl iodonium chloride (DPI, 250 µM), an inhibitor of the cell membrane NADPH oxidase-dependent ROS-synthesis that efficiently suppresses ovular ROS production. Secondly, we administered $CuCl_2$ (1 mM), a superoxide scavenger proven of strongly reducing ROS accumulation[35]. The short-term application of DPI for 15 min reduced the amount of hydroxyl radicals in the sporophytic ovule tissue and synergid region of WT and *mpk4* showing a DPI-induced ROS maximum reduction in that part of the ovule (Fig. 3p–r). Feeding *mpk4* pistils with DPI or $CuCl_2$ for 24 hrs reduced the premature synergid degeneration and significantly restored the maturation of one or both synergids compared to the mock-treated control (Fig. 3k–n). Synergid-restoration after DPI-treatment shows that NADPH oxidase-dependent ROS synthesis antagonizes synergid preservation. Likewise, $CuCl_2$ application revealed that superoxide crucially determines synergid perpetuation. Taken together, these outcomes show that MPK4-maintained ROS homeostasis in the micropylar ovule/ synergid cell region is critical for regular synergid preservation and that increased ROS levels trigger synergid PCD.

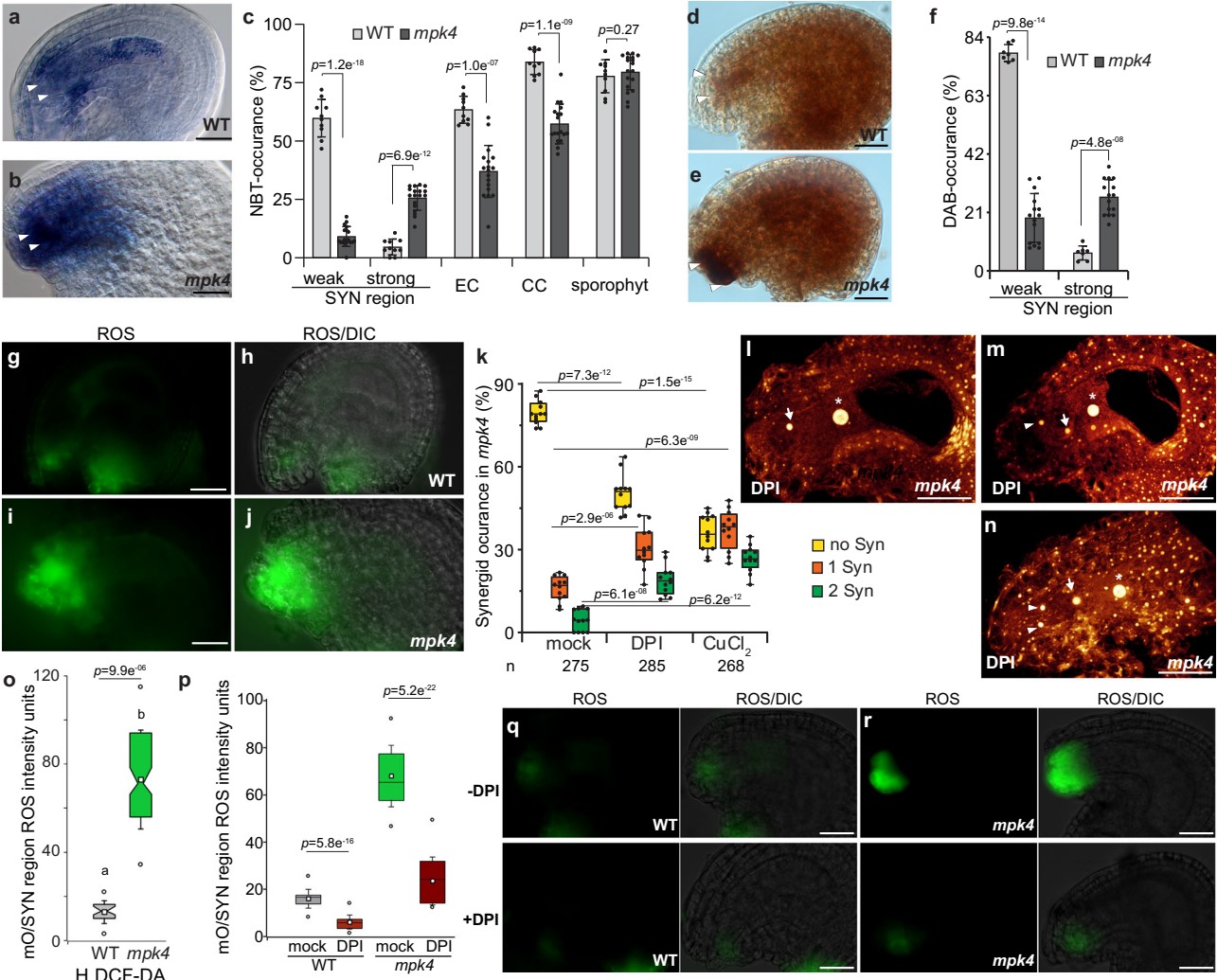

**Fig. 3 MPK4 inhibits a ROS maximum at the entrance of the female gametophyte. a–c** Superoxide detection in the sporophytic ovule tissue and the female gametophyte following nitroblue tetrazolium (NBT)-staining in WT ($n = 281$) and *mpk4* ($n = 272$), three biological replicates analyzed. **d–f** Hydrogen peroxide detection in the sporophytic ovule tissue and the female gametophyte following diaminobenzidine (DAB)-staining in WT ($n = 203$) and *mpk4* ($n = 152$), three biological replicates analyzed. **g–j, o** ROS accumulation in WT (**g, h**) and *mpk4* (**i, j**) in the micropylar ovule (mO) region shown by H$_2$DCF-DA staining. **o** Quantification of ROS-intensity units (arbitrary) in WT ($n = 283$) and *mpk4* ($n = 196$) by imageJ. Boxes represent the 25th and 75th percentiles, and the inner rectangle highlights the median, whiskers show the SD, and outliers are depicted by dots (Min/max range). mO/SYN, micropylar ovule/synergid region. **k–n** Pistil-feeding assay by the use of ROS scavengers: *mpk4* pistils of flowers at developmental stage 12b/c (Christensen et al. [22]) were treated with mock, DPI (250 µM) or CuCl$_2$ (1 mM) for 24 h and subsequently analyzed for synergid preservation by DIC analysis. Representative images after ROS scavenger-application are shown by maximum projections of Z-stacks **l–n**. Three biological replicates with a total of 275 (mock), 285 (DPI) and 268 (CuCl$_2$) *mpk4* ovules were analyzed. **p–r** Reduction of micropylar ovular ROS maximum in WT and *mpk4* by DPI-application (250 µM). Quantification of ROS-intensity units (arbitrary) in WT and *mpk4* by imageJ. Boxes represent the 25th and 75th percentiles, and the inner rectangle highlights the median, whiskers show the SD, and outliers are depicted by dots (Min/max range). mO/SYN, micropylar ovular area. Three biological replicates were performed with similar results and depicted are the results of the first replicate. Statistical significance was analyzed by one-way ANOVA. Source data and further statistical analysis are provided in the source data file. Error bars show ± SD. White arrowhead, synergid; white arrow, egg cell; white asterisk, the central cell nucleus. Scale bars, 20 µm.

**Disruption of the NB-LRR receptor *SUMM2* restores PT attraction and synergid maintenance in *mpk4*.** Under regular conditions, MPK4 suppresses the NB-LRR R protein SUMM2 that monitors the integrity of the entire MEKK1-MKK1/2-MPK4 kinase cascade. The NB-LRR receptor-mediated immune response is usually characterized by a rapid and localized cell-death, also known as hypersensitive response[29]. Thus, the auto-immunity and defense responses in *mpk4* are predominantly caused by the activation of SUMM2.

Therefore, we raised the question of whether premature synergid degeneration in *mpk4* is subjected to the SUMM2-

mediated immune-reaction. Consequently, we analyzed the *mpk4/summ2-8* double mutant that restores WT-like defense responses, hydrogen peroxide levels, and regular plant growth[16]. The hand-self-pollinated double mutant showed a high proportion of sterile seeds and aborted seeds reminiscent of the *mpk4* single mutant (Fig. 4a, b and Supplementary Fig 3h). However, we observed a small fraction of fertile seeds giving rise to the next *mpk4/summ2* homozygote generation. The fact that the double homozygote mutant generates fertile seeds indicates that the disruption of *SUMM2* in *mpk4* partly complements the *mpk4* mutant defect in seed production. Interestingly, when we

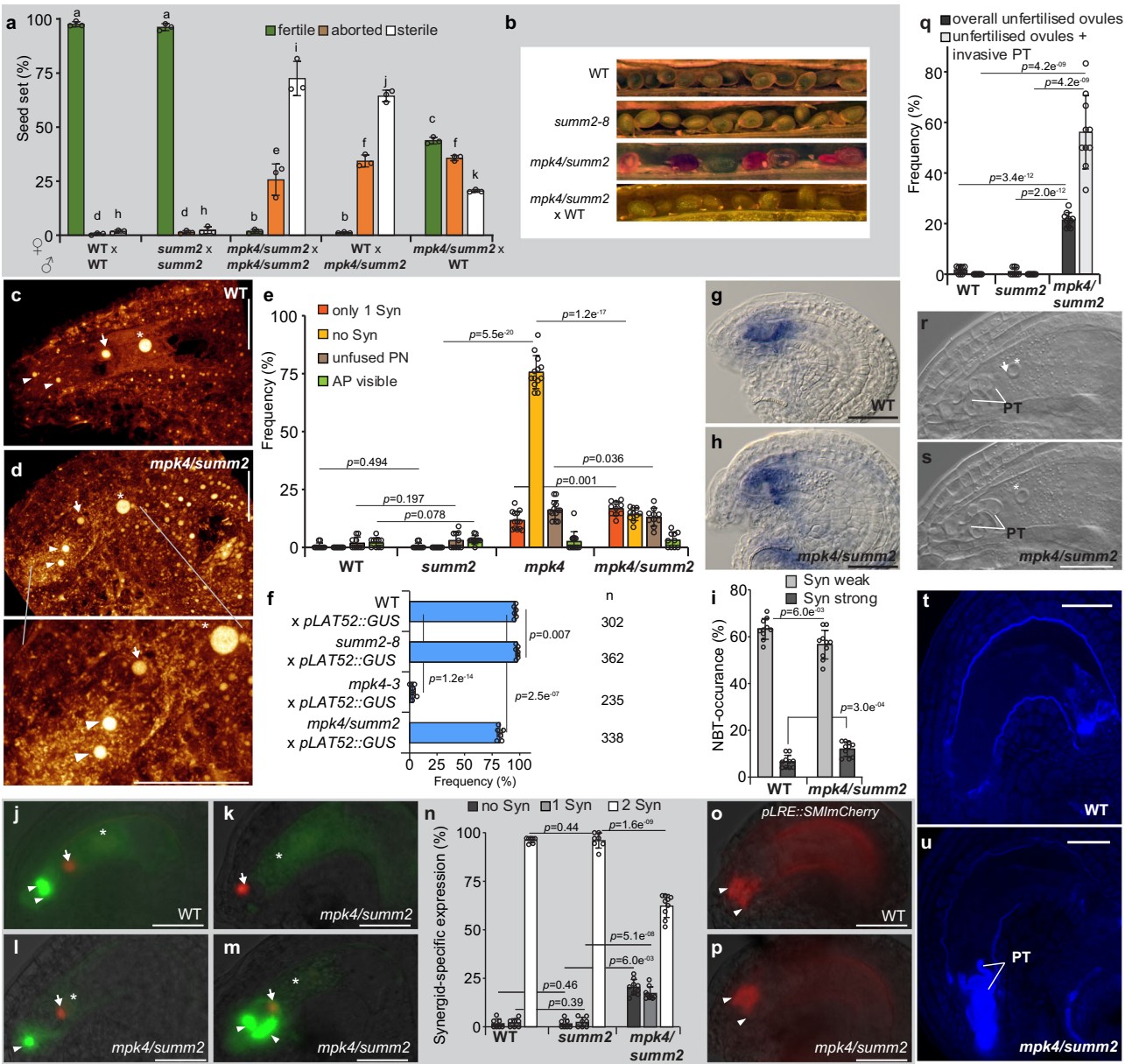

pollinated WT plants with *mpk4/summ2* originating pollen, then we observed a similar ratio of fertile, aborted, and sterile seeds as compared to the self-pollinated *mpk4/summ2* mutant. Yet, the pollination of *mpk4/summ2* with WT pollen broadly complemented the defect in the seed formation (Fig. 4a, b). Fertile seed numbers increased by up to 43% and similarly in aborted seeds, giving a total of about 79% fertilized seeds. Notably, the number of sterile seeds dropped to under 21%. These results are intriguing and indicate that the defect in seed formation in self-pollinated *mpk4/summ2* mutants is predominately caused by the *mpk4/summ2* pollen. It was reported that MPK4 is necessary for male-specific meiotic cytokinesis and MPK4-deficient mutants generate pollen bigger in size and with an aberrant number of nuclei[18]. We found that pollen of *mpk4/summ2* shows a comparable increase in size and a divergent nuclei number as in *mpk4* (Supplementary Fig. 3I). These results indicate that the disruption of *SUMM2* in *mpk4* does not rescue the male-cytokinesis defect, which strongly restricts the transmission through the male gametophyte.

Subsequently, we analyzed whether *mpk4/summ2* restores PT attraction, as a precondition for successful seed formation. We observed that the number of attracted and perceived *LAT52::GUS* PTs in the single *summ2-8* mutant[16] was indistinguishable from WT (Fig. 4f). Surprisingly, more than 80% of the *mpk4/summ2* ovules received a PT suggesting that the knock-out of *SUMM2* rescued the premature synergid-PCD observed in the *mpk4* single mutant.

To prove this assumption, we analyzed the synergid integrity in *mpk4/summ2* compared to WT, *summ2*, and *mpk4*. Compared to *mpk4*, the synergid morphology and conservation in *mpk4/summ2* are mainly restored showing that SUMM2 triggers synergid PCD in the case of MPK4 deficiency (Fig. 4c–e and Supplementary movie 6). However, we also still recognized a small portion of FGs that harbor only one or even no synergids in *mpk4/summ2* compared to WT and *summ2*. This finding indicates that the knock-out of *SUMM2* in *mpk4* does not fully complement the premature synergid degeneration in *mpk4* thereby suggesting that additional factors besides SUMM2 trigger synergid PCD in the case that MPK4 fails.

We captured in *mpk4/summ2* a weak signal for ROS accumulation in the synergids of about 57.0% and a strong

**Fig. 4 SUMM2-deficient *mpk4* plants restore synergid maintenance and function. a** Seed formation in WT ($n = 327$), *summ2-8* ($n = 439$), *mpk4/summ2* ($n = 614$), WT x *mpk4/summ2* ($n = 220$), *mpk4/summ2* x WT ($n = 428$). Error bars show ± SD. Error bars show ± SD. Statistical significance was analyzed by one-way ANOVA. Letters indicate significant differences from the mean ($p \leq 0.01$) within the same category of either fertile, aborted or sterile among the different genotypes. **b** Representative images of the seed set in the indicated genomic backgrounds. **c, d** Representative CLSM images of WT (**c**) and *mpk4/summ2* (**d**) female gametophytes 48 h post-emasculation. Depicted are maximum projections of Z-stacks, with a focal plane distance of 1 μm within a total range of about 20 μm, animated 3D-projection is provided in Supplementary movie 6. Scale bars, 30 μm. **e** Morphological characterization of the female gametophytic cells in WT ($n = 322$), *summ2-8* ($n = 323$), *mpk4-2* ($n = 340$) and *mpk4/summ2* ($n = 341$) 48 h post-emasculation. Syn, Synergids; PN, polar nuclei; AP, antipodal cells; Error bars show ± SD. **f** Pollen tube (PT) attraction assay in wild-type (WT), *summ2-8*, *mpk4-3* and *mpk4/summ2*. Plants were pollinated with pollen of *pLAT52::GUS*-expressing plants. PT attraction was analyzed 24 h after pollination. **g–i** Superoxide detection in the sporophytic ovule tissue and the female gametophyte following nitroblue tetrazolium (NBT)-staining in WT ($n = 269$) and *mpk4/summ2* ($n = 257$). **j–n** Expression of the female gametophyte reporter (*FGR7.0/+*) in WT (**j**) and *mpk4/summ2* (**k–m**). **n** Synergid-marker occurrence was determined in WT ($n = 237$), *summ2* ($n = 244$) and *mpk4/summ2* ($n = 294$) ovules. Error bars show ± SD. Scale bars, 20 μm. **o–p** Expression of the *LORELEI*-reporter (*pLRE*) coupled to a Golgi-retention peptide sequence SMImCherry. The frequency of the *pLRE::SMImCherry/-* reporter expression was analyzed in WT ($n = 338$), *summ2* ($n = 332$) and *mpk4/summ2* ($n = 307$). Statistical analysis of WT, *summ2*, and *mpk4/summ2* is shown in Supplementary Fig. 1g. **q–s** Frequency of unfertilized ovules (black bars) in WT ($n = 269$), *summ2* ($n = 262$) and *mpk4/summ2* ($n = 329$), 48 h after pollination with WT pollen. The gray bar depicts the ratio of invasive PT growth within the unfertilized ovules. In WT and *summ2*, unfertilized ovules that showed invasive PT growth could not be recognized. Error bars show ± SD. Two focal planes of an unfertilized ovule that contains an invasive PT in *mpk4/summ2* (**r, s**), 48 h after pollination with WT pollen. White arrow indicates the egg cell nucleus; white asterisk highlights the central cell nucleus; PT, pollen tube. Shown is a representative image of the results obtained from three biological replicates. **t, u** Aniline blue staining of an unfertilized ovule in WT and *mpk4/summ2* that contains an invasive PT, 48 h after pollination with WT pollen. Shown is a representative image from three biological replicates. Statistical significance was analyzed by one-way ANOVA. Source data and further statistical analysis are provided in the source data file. Error bars show ± SD. White arrowhead, synergid; white arrow, egg cell; white asterisk, the central cell nucleus. Scale bars, 20 μm (**j–m**, **g–h**, **o**, **p**, **r–u**).

accumulation of 12% (Fig. 4g–i). These values basically mirror the findings of NBT-staining in WT and are significantly restored when compared to the *mpk4* single mutant with 24.2 % strong ROS accumulation (Fig. 3c). Likewise, the NBT-staining and DAB staining at the FG4-FG5 developmental stage at the micropylar region of the female gametophyte and neighboring sporophytic tissue is moderately increased in *mpk4/summ2* compared to WT, but also significantly reduced when compared to *mpk4* (Supplementary Fig. 2a, b, e, f). After fertilization, NBT-staining of the young seeds showed weakly enhanced ROS levels in *mpk4/summ2* compared to WT and *summ2* (Supplementary Fig. 2c, d). DAB staining after fertilization resulted in a very diverse outcome from very strongly stained to almost completely unstained seeds in all analyzed genotypes (Supplementary Fig. 2g). Taken together, these results show that the introduction of *summ2* in *mpk4* partially restores ROS levels in the ovule, during gametogenesis and in the synergids. Moreover, ROS accumulation in *mpk4* is caused by SUMM2, which promotes synergid PCD.

Interestingly, we found that the expression of FGR7 in *mpk4/summ2* is distinct from the *mpk4* single mutant. In *mpk4/summ2*, the synergid marker was not detectable at all in about 20% of FGs (Fig. 4j, k, n) reminiscent of the *mpk4* mutant which exhibited the lack of this marker expression in approximately 95% (Fig. 2c). However, in *mpk4/summ2* the synergid marker was visible in a single synergid in 17% of samples and in both synergids in about 62% of FGs (Fig. 4j, l–n). Furthermore, the female gametophyte reporter *pLRE::SMImCherry* (Fig. 4o, p and Supplementary Fig. 1g) and *pHD2B::GFP:HD2B* (Supplementary Fig. 1a–d) could be detected in the majority of synergids compared to WT, *summ2* and *mpk4*.

These outcomes suggest that the premature synergid PCD in *mpk4* can be considered as an immune-related defense response triggered by the NB-LRR R protein SUMM2. Thus, SUMM2 does not only monitor the integrity of the immune MPK4-cascade concerning basal resistance, it also triggers synergid degeneration in plant reproduction.

**Impaired pollen tube rupture in *mpk4/summ2*.** Synergids attract the PT and enable the PT to rupture which results in sperm discharge. We found that synergid maintenance depends on MPK4 function. The disruption of *SUMM2* in *mpk4* largely restores synergid life-span and function. However, we found that the pollination of *mpk4/summ2* with WT pollen led to a significant number of FGs that attracted the PT but remained unfertilized and showed a lack of PT rupture (56.2%) (Fig. 4q–u). In *mpk4/summ2*, the PT continued growth within the FG, reminiscent of the *feronia* and *lorelei* phenotype[36,37], without bursting and releasing sperms. This result suggests that the lack of SUMM2 function in *mpk4/summ2* partially interferes with male-female communication and prevents synergid degeneration. Furthermore, this result implies that MPK4 does not only determines the synergid lifespan, it might also be critical for the synergid maturation to achieve complete functionality. In *mpk4/summ2*, the synergid function is restored concerning the short-track PT attraction, but mechanisms that employ PT-perception and burst might be improperly established.

## Discussion
We discovered that PCD of the synergids is governed by MPK4 which represses the nucleotide-binding leucine-rich repeat (NLR) receptor-associated cell-death reaction[16].

The receptor-like kinase (RLK) FERONIA (FER) is necessary for synergid degeneration after PT reception and exerts vital functions in plant reproduction, hormone signaling, and immunity[37–39]. The glycosylphosphatidylinositol (GPI)-anchored protein LORELEI (LRE) and LRE-like proteins (LLGs) function as co-receptors for FER. Intriguingly, in the *lre* mutant, the synergids do not collapse after PT reception reminiscent to *fer*. The filiform apparatus (FA) of WT synergids and adjacent cells in the micropylar region were found to produce higher levels of ROS in an FER/LRE dependent manner. This ROS maxima is compromised in DPI-treated pistils and in the *feronia* and *lorelei* mutant[35]. It was shown that the FER/LRE-mediated NADPH oxidase-dependent ROS accumulation in this area is crucial to set a ROS environment enabling PT rupture and sperm release after FG arrival[35]. NADPH oxidases play a pivotal role in ROS production. They transfer electrons from cytosolic NADPH or NADH to apoplastic oxygen, which leads to the production of superoxide ($O_2^-$). The produced $O_2^-$ can be converted to $H_2O_2$ by superoxide dismutase[40]. We detected high levels of various ROS

in the synergid area in *mpk4* plants showing that MPK4 represses the generation of a ROS maxima in the micropylar region of the ovule. Moreover, the application of the ROS scavenger DPI, an inhibitor of the cell membrane NADPH oxidase-dependent ROS synthesis, partially restore synergid preservation showing that synergid degeneration is intimately coupled to modulations in the ROS homeostasis. Consequently, it is conceivable that an interplay of MPK4 and FER/LRE, with their distinct function in ROS metabolism, give rise to the underlying ROS homeostasis in the micropylar embryo sac and in the FA/SC region. In the *mpk4/ summ2* double mutant, an ROS maxima is not being established thereby largely restoring synergid maturation and function.

These findings illustrate that the guardian of MPK4, SUMM2, triggers ROS accumulation and premature synergid degeneration, after being unleashed from MPK4 suppression.

PTs, fungal hyphae secrete and microbial-associated molecular patterns[41] stimulate ROS production and the generation of calcium ($Ca^{2+}$) transients[42]. ROS signaling is strongly linked to activation of several $Ca^{2+}$-permeable[42] enabling $Ca^{2+}$ flux and oscillation. Cytosolic calcium ($[Ca^{2+}]_{cyto}$) oscillations and raise are initiated in synergid cells after the physical contact with the PT apex[43]. Receptive synergids control the PCD of PTs and themselves by coordinating their distinct calcium signatures in response to the calcium dynamics and growth behavior of the PT[44]. FER and LRE pathways are required for calcium oscillation in the synergids, which couples the PCD events in the receptive synergid and the PT[44]. Thus, it is conceivable that the $[Ca^{2+}]_{cyto}$ increase and dynamics in the receptive synergid might interfere with MPK4 functions and might trigger synergid PCD upon SUMM2 activation.

Recently, it was shown that FER and the closely related RLKs, LET1 and LET2, form a specific trimeric *Catharanthus roseus* RLK1-like (CrRLK1L) module together with the GPI-anchored LLG1 to modulate SUMM2-mediated autoimmunity[45]. Sensing of cell wall integrity by CrRLK1L-associated pathways has been uncovered as being eminent for cell homeostasis regulating plant immunity, growth, and reproduction[46]. It is plausible that a similar CrRLK1L complex modulates SUMM2-mediated cell death in the synergids in response to PT-reception. In this respect, PT reception in synergid cells depends on the CrRLK1Ls: HERCULES RECEPTOR KINASE 1 (HERK1) and ANJEA (ANJ)[47], that interact with LRE and FER, in a proposed trimeric signaling complex. The proposed FER, LRE, HERK1/ANJ trimeric complex could be the synergid cell counterpart to the immunity FER, LLG1, LET1, LET2 complex thereby modulating SUMM2-mediated ROS accumulation and PT-perception.

In contrast to the *summ2* single mutant, we found a PT-overgrowth phenotype in *mpk4/summ2*, reminiscent of *fer, lre, nortia (nta)*[38] and *herk1/anjea*. The disruption of *MPK4* in *summ2* might compromise FER, LRE, NTA or HERK1/ANJEA and eventually the function of the trimeric complex. This effect might not be observable in the *mpk4* single mutant owing to the predominant absence of PT-perception. NTA-GFP is homogeneously distributed in synergids in compartments that co-localizes with a *cis*-Golgi marker[48]. After PT arrival, NTA-GFP is solely detected in the FA. In *fer* and *lre*, NTA-GFP is not translocated to the FA showing that NTA movement and accumulation to the FA depends on FER/LRE function. Yet, forced localization of a chimeric NTA-version (faNTA) in the FA suppresses the *fer/lre* PT reception phenotype illustrating that NTA accumulation at the FA can bypass the FER/LRE pathway[49]. We showed that the premature synergid degeneration in *mpk4* was partly complemented by introducing *summ2*. However, to some extend the synergids are still compromised and mechanisms that employ PT-perception might be improperly established, which

might be reflected by an unusual distribution of NTA after PT reception. The NTA distribution towards the FA plasma membrane might still be perturbed in *mpk4/summ2* resulting in an improper PT-reception reminiscent to the phenotype obtained in mutations of the proposed trimeric complex and *nta*.

Moreover, introducing *mpk4* in *summ2* might affect additional guard proteins or regulatory factors that interfere with synergid function and PT-reception. In single *summ2*, redundant factors, whose function depends on MPK4, might compensate for the lack of SUMM2. Following this notion, disruption of the TNL immune receptor SMN1/RPS6[50,51] and of the DEAD-Box RNA Helicase SMN2/HEN2 were shown to widely restore *mpk4*[50], reminiscent to *mpk4/summ2*. Mutations in *SMN1/RPS6* and *SMN2/HEN2* partially suppressed ROS-accumulation and the dwarf, and autoimmune phenotype of *mpk4* plants. Thus, the structurally distinct NLR proteins, SMN1/RPS6, SMN2/HEN2 and SUMM2, monitor the integrity of the MPK4 pathway and might exert a distinct impact on PT-perception following the disruption of MPK4. The proper functionality of the MPK4-cascade is guarded by SUMM2 and other NBS-LRRs to hinder infection progression. However, MPK4 also exerts multiple functions in the cell that are distinct from the SUMM2-triggered cell death[12]. Thus, in *mpk4*, two types of defects emerge, those caused by the direct lack of MPK4 in the cell, and those of consequence of SUMM2/NBS-LRRs triggered cell death. Therefore, only *mpk4* defects that cannot be complemented by mutations of the NBS-LRRs hold potential for unraveling MPK4-specific functions in the plant. In this regard, the pollen tube overgrowth in *mpk4/summ2* might refer to a MPK4-specific SUMM2-independent function in plant reproduction. Since MPK4 is expressed both in synergids and integuments, and the SUMM2-induced reproductive impairment is of sporophytic origin, it is possible that MPK4 is critical for FER/LRE/NTA signaling within the synergids, and is guarded by SUMM2 in the integuments as a safekeeping mechanism. Eventually, the precise mechanistic reason for this phenotype remains to be determined, which would be worth tackling in further studies.

Given that both synergids prematurely undergo cell death in the *mpk4* mutant, we reasoned that MPK4 maintained the synergid function in both, the receptive and persistent synergid. Thus, the molecular organization triggering PCD in the synergids seems to be conserved, supporting the notion of a uniform synergid fate before PT reception, that is not per se subdivided into either receptive and persistent synergid cell fate.

We found that the synergid pre-maturation defect in *mpk4* homozygote mutants is of sporophytic origin and originates from the high ROS accumulation in the predominantly sporophytic micropylar region of the embryo sac right in front of the FG's synergids. Mutations in the highly glycosylated ARABINOGALACTAN PROTEIN 4 (AGP4)/ JAGGER affects the persistent synergid's cell death after fertilization and results in polytubey[52]. JAGGER locates to the cell surface and its sugar moieties have been implicated with the signaling process, as described for the glycosides of the PT activator AMOR[19,53]. Interestingly, JAGGER is a sporophytic factor and the survival of the persistent synergid after fertilization is also consequently of sporophytic origin in *jagger* (−/−). The sporophytic origin of their mutant phenotypes, demonstrates direct control of the maternal ovule tissue over the maturation and function of the synergids and consequently on seed production.

JAGGER is deemed to be downstream of the ethylene (ET)-signaling pathway[52] which was shown to exclusively govern the disintegration of the persistent synergid. In the ET hyposensitive mutants *ein2* and *ein3*, the persistent synergid is maintained after fertilization, and additional PTs are attracted[23]. Moreover, EIN3-

mediated synergid nuclei disintegration triggers endosperm expansion[54]. Thus, it is conclusive that two different underlying mechanisms coordinate the degeneration/disintegration of the synergids. On one hand, the guardian of MPK4, SUMM2, induces ROS-triggered cell death in the PT-receipting synergid after the PT-arrival at the ovule. On the other hand, the ethylene-signaling cascade coordinates the disintegration of the persistent synergid in the course of gamete fertilization and seed expansion.

The premature degeneration of the synergids in a SUMM2-dependent manner might refer to an underlying molecular mechanism for the suppression of plant reproduction in the course of systemic NLR-mediated immunity. It is imaginable that acquiring resources beneficial for NLR-triggered defense response, called effector-triggered immunity[55], might reasonably be at the expense of seed formation. Following this logic, to negotiate and finally reallocate resources from plant reproduction towards immunity, a tight control over the synergid function might be critical to suspend PT reception, sperm-cell release, and seed formation

Reliable plant reproduction and sustainable defense response to pathogens are the two main cost-intensive processes ensuring plant prosperity and species survival. Considering concurrent seed production and pathogen resistance in agriculture, deciphering underlying signal transduction pathways, that balance both processes, will be beneficial for future food resources.

## Methods

**Plant material and growth conditions**. Arabidopsis plants (Columbia accession) were grown on soil and sugar-free MS-plates under long-day conditions (16 h light: 8 h dark) at 24 °C. Seeds of *mpk4-2* (N556245), *mpk4-3* (EMS-mutagenized, G186E exchange), *summ2-8* (N842411) and *mpk4/summ2-8*, FGR7.0, *pMPK4::MPK4:GFP*, *pHB2D::GFP:HB2D* were received on request by the corresponding authors of the indicated citations. *pLAT52::GUS* (N16336), *pAGL62::AGL62-GFP-GUS* (N799498), *pLRE::GFP* (N69973), *pLRE::SMI:mCherry* (N69715) containing seeds were obtained from NASC.

**Histology and microscopy**. For the analysis of the female gametophyte by differential-contrast (DIC) microscopy and confocal-laser scanning microscopy (CLSM), we followed the instructions given by Völz et al.[21]. The oldest closed flower bud was emasculated and analyzed 48 h later. The flower was cleared in Corney's solution (9:1 ratio of 100% ethanol and acetic acid) for 24 h. Subsequently, rehydration of the sample was performed by using 80% and 70% ethanol for a half-hour each. The pistil was separated from the flower and mounted on an object slide in 40 µl visikol (optical clearing agent). Afterward, the pistil was opened by the use of fine needles to release the ovules. For CLSM observation, the pistil's replum was slit opened on both sides using an injection needle and incubated in fixation buffer: 4% glutaraldehyde, 12.5 mM cacodylate buffer, pH 6.9 for 4 h. After dehydration in an ethanol dilution series (10, 20, 40, 60, 80, 95, and 100%), the sample was cleared in a 2:1 ratio of benzyl benzoate: benzyl alcohol for 15 min. The samples were mounted in immersion oil and sealed under a cover slide by the use of colorless nail polish.

Samples expressing GFP, YFP, dsRED, and dtTomato were mounted on 7.5% glycerol and observed on a ZEISS epi-fluorescence microscope. CLSM was performed on the Leica Confocal Laser Scanning Microscope SP8 with an excitation wavelength of 488 and 568 nm laser line.

**Histochemical reactive oxygen species detection**. In situ detection of $H_2O_2$ was performed with the use of 3,3'-diaminobenzidine (DAB). The pistils of flowers 24 h post-emasculation, were laterally opened and submerged in DAB solution (50 mg DAB, 130 mg $Na_2HPO_4$, 0.01%v/v Tween 20) and incubated for 4 h. Subsequently, the pistils were mounted in 20% glycerol. The staining of superoxide radicals was carried out by the use of Nitroblue tetrazolium (NBT). The laterally opened pistils were incubated in NBT-solution of 30 min at 37 °C and subsequently mounted in 20% glycerol. For $H_2DCF-DA$-detected ROS accumulation, pistils were carefully laterally opened by the use of a fine needle and immersed in 10 µM $H_2DCF-DA$ in ROS-staining buffer (50 mM $CaCl_2$, 5 mM KCl, 10 mM MES, pH 6.15) at room temperature for 5 min. Pistils were washed in ROS-staining buffer for 2 min. Ovules were released and mounted on an object slide in ROS-staining buffer, covered by a cover slide, and observed at a ZEISS epi-fluorescence microscope. ImageJ was used to determine intensity and threshold values.

**Pistil-feeding assay for synergid restoration**. The oldest closed flower bud was transferred and planted into 1 ml solidified 0.7% agarose in a 1.5 ml Eppendorf vial.

Afterward, the vial was closed and stored at long-day conditions at 24 °C. The agarose was either supplemented with DPI (250 µM), to detect NADPH oxidase-dependent ROS production or the superoxide scavenger $CuCl_2$ (1 mM). After 24 h, the flowers were incubated in Corney's solution for DIC-microscopy or fixation buffer for CLSM observation.

**Semi-in vivo pollen tube attraction and calcofluor white staining**. 1 ml of semi-in vivo pollen tube medium (0.01% $H_3BO_3$, 5 mM $CaCl_2$, 5 mM KCl, 1 mM $MgSO_4$, 15% sucrose, adjusted pH to 7.5 with KOH) was supplemented with 0.012 low-melting agar and carefully heated up in a microwave. 80 µl were spread on an object slide and stored in a moisture box to avoid dehydration of the media. WT stigmas, of flowers 48 h after emasculation, were decapitated and put on the solid growth media. Four to six ovules were placed in front of the excised stigma. Subsequently, the stigma was pollinated with WT pollen. Six hours after pollination the pollen tube attraction rate was determined. Calcofluor white (fluorescence brightener 28, 0.01%w/v) solution plus induction-solution (10% KOH, 10% glycerin, 1:1 v/v) were applied on the semi-in vivo pollen tube approach and analyzed after 10 min by epi-fluorescence microscopy.

**GUS staining**. Emasculated flowers were pollinated with *pLAT52:GUS*-carrying pollen. 24 h later, the pistils were slit opened to remove the carpel walls and transferred to GUS staining solution (10 mM EDTA, 0.1% Triton X-100, 2 mM $K_4Fe(CN)_6$, 2 mM $K_3Fe(CN)_6$, and 1 mg/mL 5-bromo-4-chloro-3-indolyl glucuronide (X-Gluc) in 50 mM sodium phosphate buffer, pH 7.2). A vacuum was applied for 30 min and the pistils were subsequently incubated at 37 °C for 24 h. The pistils were dissected and taken up in 80 µl of 80% glycerol, covered by a coverslip, and inspected under a light microscope.

**Aniline blue staining**. Pistils were cleared in visikol (optical clearing agent) at 65 °C for 5 min, washed with water, and softened with 5 M NaOH at 65 °C for 5 min. The pistils were washed with water and incubated in aniline blue staining buffer (0.1% aniline blue in 0.1 M $K_3PO_4$ buffer, pH 8.3) for 3 h in the dark. Afterward, the pistils were washed with 0.1 M $K_3PO_4$ buffer and analyzed in a drop of glycerol by CLSM.

**Statistical analysis**. Statistical significance was calculated based on one-way ANOVA. Different letters above bars indicate significant differences, $p \leq 0.01$. Samples sharing letters are not significantly different. Error bars show ± Standard-deviation of the mean (SD). Three biological replicates (BR) were performed and shown is the average of three BGs, or a representative result of one BG (see also source data file).

**Transmission efficiency calculation ($TE_F$)**.

$$Te_F(\%) = (\text{heterozygote mutant progeny}/\text{wild type progeny}) \times 100$$

**Reporting summary**. Further information on research design is available in the Nature Research Reporting Summary linked to this article.

## Data availability

All data generated or analyzed during this study are included in this published article (and its supplementary information files). Source data are provided with this paper.

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

## Acknowledgements
This work was supported by the National Research Foundation of Korea (NRF) grants funded by the Korea government (MSIT) (2020R1A2B5B03096402, 2018R1A5A1023599, and 2021M3H9A1096935). WH is grateful for a graduate fellowship through the Brain Korea 21 Plus Program.

## Author contributions
R.V., H.H., and Y.H.L. designed and analyzed experiments. R.V. and W.H. performed the experiments. R.V. wrote the original draft. H.H., W.H., and Y.H.L. edited the manuscript.

## Competing interests
The authors declare no competing interests.
