## [Peer Review File · Nature Communications]

ROS homeostasis mediated by MPK4 and SUMM2 determines synergid cell deathREVIEWER COMMENTS

Reviewer #1 (Remarks to the Author):

In *Arabidopsis*, two synergid cells in the female gametophyte are responsible for attracting the pollen tube to grow into the micropyle, and two sperm cells are released after pollen tube rupture. Then two sperm cells fuse with the central cell or the egg cell, forming the endosperm or the embryo respectively. The receptive synergid undergoes programmed cell death (PCD) instantly during pollen tube reception and rupture, whereas the persistent synergid will stay intact until the second fertilization is completed. Previous studies have revealed that synergid degeneration occurs as a PCD response, and ROS is well known to trigger PCD in plant cells. In plant immune response, the immune-activated MAP kinase 4 (MPK4) functions as a suppressor of PCD, by suppressing the NB-LRR R protein SUMM2. In this study, Völz et al. reported that MPK4 also suppressed SUMM2-mediated PCD by regulating ROS homeostasis and as a result synergid PCD is prevented. In *mpk4*, ROS levels are increased and synergid cells prematurely undergo PCD before PT reception. However, either treatment with ROS scavengers or genetic disruption of SUMM2 in the *mpk4* background restored ROS homeostasis, synergid maintenance and PT perception, suggesting that MPK4 and SUMM2 are suppressors for synergid PCD. In *mpk4 summ2*, PTs show a *feronia*-like overgrowth phenotype, revealing SUMM2-mediated ROS functions in pollen tube rupture. Based on these results, the authors conclude that synergid cell death is suppressed by MPK4 that suppresses SUMM2-mediated maintenance of ROS homeostasis. The work is interesting to the readers in the plant field, and the genetic evidence is good. However, I still have some concerns that need to be addressed before the paper can be accepted by Nature Communications.

Major points:

1. The authors will need to demonstrate that the development of the female gametes in *mpk4* is normal, the only problem is that the synergids prematurely undergo PCD. The development of female gametes at different FGs can be observed with three methods, CSLM, DIC and fluorescence observation. Meanwhile, there should be statistically analyzed. For example, in Fig 1T, the FG6 value of WT should theoretically be close to 50%, but here in this study, it is only 30%. Need reasonable explanations for the missing 20%.
2. What is the expression pattern of MPK4 in the different FG stages? When does the expression of MPK4 start and end? In Fig 2Q, MPK4 can be detected in the synergid nuclei, egg, and CC nuclei. Since MPK4 is not specifically expressed in the synergids, it is necessary to use a synergid cell-specific promoter (e.g., the MYB98 promoter) to drive MPK4 for genetic complementation.
3. In Fig 2O, the cell pointed by the white arrowhead doesn't look like a synergid cell, is it possible that they are not complemented? Where are the results of DIC observation, as referred in Fig 2M-2P legend? More detailed data need to be provided to show that DPI or CuCl₂ treatment can really help to preserve the synergids.
4. Fig 2Q/2S are ambiguous and not clear. Firstly, the resolution of Fig 2Q/2S need to be increased. Secondly, it'd better to distinguish the receptive from persistent synergid. The authors claimed that MPK4 exhibited a distinct localization in the synergids after pollen tube reception, but could it possibly be an artifact from the PCD of the receptive synergid? In order to rule out this possibility, I recommend the authors to use female gametophyte cell/nuclear reporter lines to observe more closely.
5. The ROS staining results at different FG stages should be provided, which will help the readers see clearly that the abnormal level of ROS starts from FG7.
6. Fig 4 shows that pollen tube rupture is compromised in *mpk4 summ2*, but there is no direct evidence that the phenotype is related to ROS. The author should compare the ROS levels in ovules of *mpk4*, WT, and *mpk4 summ2* before and after pollination.
7. The conclusion in Line123 is not accurate.

8. Model in Fig 4F needs to be revised. The position of PT rupture is incorrect and it is not right to draw two synergids, endosperm and embryo appear at the same time. The model can be presented separately, according to before and after receiving pollen tube reception.
9. Overall, the quality of the pictures in the article needs to be improved (Fig 2O, 2Q, 2S), and the statistical analysis needs to be more rigorous (Fig 2R).

Minor points :

1. Is Fig 1K the result of CSLM or DIC? Please indicate clearly.
2. Why does the cell marked by red fluorescence in Fig 1M look obviously larger than that in Fig 1L? Are the ovules of Fig 1L and 1M at the same developmental stage? I can see the sizes of the ovules are different. Please clarify it.
3. The NBT-occurrence (%) of EC in *mpk4* is lower compared to WT in Fig 2D, but the results of NBT staining in Fig 2B/2D seemed different from statistical results.
4. Is Fig 2M-2O the result of DPI or CuCl₂ treatment?
5. In Line 161, 2C-2D should be changed to 2B-2D.
6. In the Legend of Fig 3J-M, please clarify what the white arrowhead or white asterisk means.
7. Some figures are not properly referred in the text, such as Fig 2M-P and Fig 4F.

Reviewer #2 (Remarks to the Author):

The manuscript submitted by Völz et al. describes the MPK4 and SUMM2 signalling module in the development of the female gametophyte in *Arabidopsis thaliana*. While the infertility of *mpk4* homozygous plants has been previously described, and the role of MPK4 in pollen development is now better understood, little is known about this protein's activity in developing female gametophytes.

The authors conduct a series of experiments to structurally characterise the female gametophyte development and pollen tube reception in *mpk4* mutants. The experiments unequivocally show the *mpk4*(-) synergid cells undergo premature degeneration that also correlates with increased ROS accumulation. Given the well-established role for MPK4 in suppressing autoimmunity, ROS production and PCD under pathogenically unchallenged conditions, the authors hypothesise a similar role for MPK4 in synergid cells. Next, the authors test whether mutating the NB-LRR SUMM2 in an *mpk4* background can rescue the reproductive defects of *mpk4* plants. While the pollen defects seen in *mpk4* plants were still present in *mpk4 summ2* double mutants, the female gametophyte and reproductive defects are lessened in *mpk4 summ2* ovaries, suggesting a role for SUMM2 as a guard for MPK4 downstream targets in synergid cells, similar to what has been described for this signalling module in defence. Finally, Völz et al. identify an invasive pollen tube phenotype in *mpk4 summ2* ovules, reminiscent of the *fer* and *Ire* phenotypes.

The work presented in this manuscript will be of interest to the plant science community as it clarifies the role of MPK4 in the female gametophyte and evidences the parallels between defence and pollen tube reception mechanisms in *Arabidopsis*. The manuscript is well written and the figures well presented. The methodology and statistical analyses also seem correct. This reviewer nevertheless has a major concern with this study that the authors should experimentally address to substantiate most of their claims: demonstrating the reproductive defects in *mpk4*(-/-) ovaries is due to defective synergid cells. It is highly likely that the *mpk4*(-/-) ovaries may produce high levels of ROS in other maternal tissues that would impair the pollen tube journey towards the ovule. For this reason, the authors should carry out a series of experiments with hemizygous *mpk4*(+/-) ovaries to demonstrate the gametophytic origin of the *mpk4* maternal reproductive defects.

Below, I have listed the revisions that this reviewer believes should be conducted to make of this manuscript worthy of publication.

Major revisions that should be addressed to provide solid evidence to support the author's claims:

Line 81 – High rates of untargeted ovules could also originate from problems in pollen grain hydration and germination at the stigma, a problem in pollen tube growth through carpel tissues, etc. Authors should show whole GUS-stained ovaries. If mpk4 is exclusively affecting PT attraction from the synergids, the carpels will show fully grown pollen tubes all along the transmitting tract that are not growing into the ovules. If that's not the case, the authors should describe at which particular step along the PT journey the defects arise.

Line 113 - Since mpk4 homozygous mutant displays widespread deleterious phenotypes, before drawing any conclusions, it is key to demonstrate that the synergid cell defects are gametophytic, and rule out a sporophytic effect caused by surrounding maternal tissues. The authors should provide an extra figure in which they show detailed analysis of the expression pattern of pMPK4 in the ovules, at the two stages they have analysed (FG5, FG7) grown and emasculated under their same conditions. A nuclear-localised fluorescent marker should be used for this experiment to provide the cellular resolution needed within the ovule. Secondly, the authors need to show the gametophytic effect of mpk4 mutation. Repeat the pollination and GUS staining experiment with mpk4(+/-) ovaries, in which they should see around 50% of ovules with PT reception (mpk4(+)) female gametophytes, and 50% without (mpk4(-)) female gametophytes. And finally, repeat the synergid cell integrity analysis in an mpk4(+/-) background, either by CLSM of fixed ovules, or simply with one of their synergid-specific fluorescent markers, in which, again, they should register approximately a 50% increase in the ovules displaying clear synergid cell signal 48h post emasculation. If the experiment yields different results, the authors should reinterpret their results and change their conclusions and discussions accordingly. In addition, the authors could complement the mpk4(-/-) mutant with a synergid cell-specific promoter construct (LREpro:MPK4, for example) to demonstrate that the reproductive defect stems from the synergids and not from other sporophytic tissues. If running these experiments reveal that MPK4 is actually controlling pollen tube germination or growth through stigma and style, the authors should then test in semi-in vitro fertilisation experiments whether pollen tubes germinated and grown through WT stigma and styles are attracted by exposed mpk4(-/-) ovules.

Line 118 - Pollinating with WT does not preclude pollen tubes being unable to reach the ovules in mpk4 pollinations, it simply provides fully functional pollen grains. Again, the maternal tissues in mpk4 plants (stigma, style, transmitting tract, etc.) could have a direct impact on the pollen's fertilisation success. The authors need to show the GUS assay in whole carpels so that it is possible to observe at which stage the pollen is failing to reach the ovules.

Line 217 – Images in Figure 2Q-S are not clear enough. Authors need to provide confocal images from the ovules expressing MPK4-GFP. And authors should not forget to image in parallel a WT control to compare the autofluorescence in the ovular tissues, that is generally pretty obvious and can be misleading when analysing the MPK4-GFP signal, especially if the synergids undergo degradation.

Revisions that should be addressed to improve the quality of the presented results and discussion:

Line 259 - In this experiment (Figure 3F), the control should have included mpk4 single mutant as analysed in Figure 1.

Line 380 - How can the authors argue that the MPK4-SUMM2 module is involved in PT reception when the SUMM2 single mutant has a wild-type PT reception phenotype? If the synergid CrRLK1Ls signalled via MPK4-SUMM2 to trigger synergid cell degeneration after PTs arrive, then summ2 single mutants should display a PT overgrowth phenotype, as well as the mpk4/summ2 double. Authors should address this issue in the discussion.

Line 395 - The results presented by the authors do not point at the connection between a pathogenic infection, and MPK4-SUMM2 reducing fertility as a response. Such claims may be misleading as the authors presented no evidence supporting such connection between systemic resistance and fertility decrease via MPK4. What the authors are presenting is rather a story of how signalling cascades can be co-opted in different contexts, to trigger PCD upon pathogens, or to trigger PCD in ovules in synergid cells upon PT reception. I would suggest the authors rather discuss the connection between pathogen responses and pollen reception (ROS production, Ca²⁺ signalling, FERONIA and LLG proteins, NORTIA, and now MPK4 and SUMM2) from an evolutionary standpoint.

Minor revisions/comments:

Line 37 - This describes Arabidopsis thaliana female gametophyte development, but not many other angiosperms or other types of plants. Please rephrase.

Line 43 - Sentence needs rewriting - missing verb

Line 55 - What is male-cytokinesis? rephrase to give more detail about the described role of MPK4 in pollen development

Line 56 - In which pathway does this occur? in the reproductive context? In a defence pathway? Be more precise to aid reader understand the background. MPK4 doesn't necessarily have to interact with the same signalling partners in different developmental contexts.

Line 93 - Rephrase "joins together" for better understanding. I suggest: "This marker allows simultaneous characterisation of synergid cell, egg cell, and CC in a single construct".

Line 116 - Why is this panel in Figure 2 that focuses on ROS production? would be better fitting as part of the general description of the phenotype of mpk4 in Figure 1. Or split Figure 1 into two so that it does not contain so many panels.

Line 123 - Authors have to add solid evidence to substantiate this claim. At the moment it is speculation.

Line 161 - Wording: intense, instead of intensive.

Line 171 - The images provided are not of enough quality/resolution to say the staining signal comes from FA or not. Authors should rephrase and say there is strong signal at the micropylar region.

Line 259 - This does not necessarily suggest anything about the synergids in particular, since it has not been shown that the pollen tubes are reaching the ovules normally in mpk4

ovaries. SUMM2 could be reducing autoimmunity phenotypes along the carpel tissues, and therefore allowing pollen tubes to germinate and grow through them and be attracted towards the ovules.

Line 354 - Authors have not discovered that. Authors have discovered MPK4 and SUMM2 affect synergid cell integrity and possibly the pollen tube reception process. The authors have not provided any evidence for the additional MAPK cascade elements in the synergids – please rephrase to avoid misleading the reader.

Line 379 - FER is a CrRLK1L. I suggest re-writing into: "It is conceivable that a similar CrRLK1L complex modulates SUMM2-mediated cell death in the synergids in response to PT-reception". PT reception in synergid cells depends on other CrRLK1Ls: HERK1 and ANJ, that interact with LRE and FER, in a proposed trimeric signalling complex. The authors should comment that such FER, LRE, HERK1/ANJ trimeric complex could be the synergid cell counterpart to the immunity FER, LRG1, LET1, LET2 complex.

Line 407 - Please indicate the T-DNA stock IDs for mpk4 and summ2 lines.

Line 515 – Figure S1 should be split into two Supp figures. First one regarding the genetic reporters, and the second regarding the seed set and pollen size

Firstly, many thank the editor and reviewers for investing their time and efforts in considering our manuscript. We are very grateful for the constructive suggestions and advice.

As detailed in the point-by-point response to every comment below, we have performed additional experiments and analyses that consolidate our findings and claims on the revised manuscript.

We hope you share our enthusiasm for this project and we wish you and your family to be safe and sound through the CoVID19-pandemic.

REVIEWER COMMENTS

Reviewer #1 (Remarks to the Author):

In Arabidopsis, two synergid cells in the female gametophyte are responsible for attracting the pollen tube to grow into the micropyle, and two sperm cells are released after pollen tube rupture. Then two sperm cells fuse with the central cell or the egg cell, forming the endosperm or the embryo respectively. The receptive synergid undergoes programmed cell death (PCD) instantly during pollen tube reception and rupture, whereas the persistent synergid will stay intact until the second fertilization is completed. Previous studies have revealed that synergid degeneration occurs as a PCD response, and ROS is well known to trigger PCD in plant cells. In plant immune response, the immune-activated MAP kinase 4 (MPK4) functions as a suppressor of PCD, by suppressing the NB-LRR R protein SUMM2. In this study, Völz et al. reported that MPK4 also suppressed SUMM2-mediated PCD by regulating ROS homeostasis and as a result synergid PCD is prevented. In *mpk4*, ROS levels are increased and synergid cells prematurely undergo PCD before PT reception. However, either treatment with ROS scavengers or genetic disruption of SUMM2 in the *mpk4* background restored ROS homeostasis, synergid maintenance and PT perception, suggesting that MPK4 and SUMM2 are suppressors for synergid PCD. In *mpk4 summ2*, PTs show a *feronia*-like overgrowth phenotype, revealing SUMM2-mediated ROS functions in pollen tube rupture. Based on these results, the authors conclude that synergid cell death is suppressed by MPK4 that suppresses SUMM2-mediated maintenance of ROS homeostasis. The work is interesting to the readers in the plant field, and the genetic evidence is good. However, I still have some concerns that need to be addressed before the paper can be accepted by Nature Communications.

Major points:

1. The authors will need to demonstrate that the development of the female gametes in *mpk4* is normal, the only problem is that the synergids prematurely undergo PCD. The development of female gametes at different FGs can be observed with three methods, CSLM, DIC and fluorescence observation.

We found that the female gametes in *mpk4* exhibited a cell specific marker expression that was indistinguishable from WT (**Fig.2A-C**). The *mpk4* egg cells expressed the *EC1* reporter, previously shown to mark egg cell fate, and the central cell expresses the *DD22* marker which has been well established as a CC-marker in various publications [1, 2]. To analyze whether the molecular profile of the *mpk4* gametes is in accordance with their function, we analyzed whether embryo and endosperm are formed after WT pollination. Despite the very low pollen tube perception rate in *mpk4*, we could confirm that in the rare cases, when a PT indeed has delivered the sperm cells, then an embryo and endosperm was formed in *mpk4*. (**S-Fig. 3G**) Thus, the molecular and functional analysis of the *mpk4* gametes indicate that they are not broadly perturbed.

Meanwhile, there should be statistically analyzed. For example, in Fig 1T, the FG6 value of WT should theoretically be close to 50%, but here in this study, it is only 30%. Need reasonable explanations for the missing 20%.

Thank you for making us aware of that inaccuracy, we added a stringent statistical analysis to every analysis.

We analyzed the oldest closed flower bud of *mpk4* and WT concerning synergid establishment, which corresponds to the developmental stage FG5, but not FG6, as mistakenly indicated in the previous Fig.1 T. At

the FG5 stage the cell fates within the FGs are about of being established, thus the occurrence of the synergid marker is below 50%. We agree with the reviewer that at the FG6 stage, the expected rate of synergids showing a GFP signal should be at approximately 50%.

2. What is the expression pattern of MPK4 in the different FG stages? When does the expression of MPK4 start and end? In Fig 2Q, MPK4 can be detected in the synergid nuclei, egg, and CC nuclei.

We detected MPK4:GFP driven by *pMPK4* after the cellularization of the 8 nuclei-containing syncytium within the FG at developmental stage FG5-FG6. We detected MPK4:GFP in the synergids, egg cell and central cell nucleus, as well as in the cytoplasm (Fig. 2M-P). Additionally, we found a strong expression of MPK4 in the ovular tissue that surrounds the FG (Fig. 2K-L).

Since MPK4 is not specifically expressed in the synergids, it is necessary to use a synergid cell-specific promoter (e.g., the MYB98 promoter) to drive MPK4 for genetic complementation.

To reveal the inheritance pattern of *mpk4*, we performed reciprocal crosses with WT plants (Fig.2J) and determined the female and male transmission efficiency (TE_f). The TE_f was close to 100% for the transmission of *mpk4* through the female and male gametophyte which indicates a sporophytic origin of the synergid prematuration defect in *mpk4* homozygote mutants. The transmission rate of the self-crossed *mpk4* heterozygote mutant basically mirrors a Mendelian segregation of 1:2:1 (S-Fig. 3F). Furthermore, we showed that the *mpk4* heterozygote mutant is not affected concerning the premature synergid degeneration, seed formation and synergid-marker expression (Fig. 1Q, S-Fig. 3A-E). Thus, we feel the complementation of the *mpk4* mutant by a synergid-specific expression of a rescue-construct is not required at this stage of experimental analysis and we would like to refrain from expanding into this direction as it seems outside of the framework of this manuscript.

3. In Fig 2O, the cell pointed by the white arrowhead doesn't look like a synergid cell, is it possible that they are not complemented? Where are the results of DIC observation, as referred in Fig 2M-2P legend? More detailed data need to be provided to show that DPI or CuCl₂ treatment can really help to preserve the synergids.

We would like to apologize for the insufficient image resolution. We replaced the images showing the restoration of the *mpk4* FG after DPI-treatment. Representative images were taken by CLSM, and are presented as maximum projections of a Z-stack observations. The counting was performed by analyzing samples after DIC observation (Fig. 3K-N).

Furthermore, to evaluate further whether these ROS scavengers can affect the strong ROS accumulation in *mpk4*, we performed a H₂DCF-DA staining after a short-term DPI treatment. As introduced by Duan et al, 2014 for WT plants, we found that DPI-treatment reduces the ROS accumulation in the micropylar region of the ovule in *mpk4* compared to WT. Thus, the restoration of synergid maturation after DPI/ CuCl₂ treatment can be traced back to the reduction of the ROS level after ROS scavengers' application (Fig. 3P-R).

4. Fig 2Q/2S are ambiguous and not clear. Firstly, the resolution of Fig 2Q/2S need to be increased. Secondly, it'd better to distinguish the receptive from persistent synergid. The authors claimed that MPK4 exhibited a distinct localization in the synergids after pollen tube reception, but could it possibly be an artifact from the PCD of the receptive synergid? In order to rule out this possibility, I recommend the authors to use female gametophyte cell/nuclear reporter lines to observe more closely.

We reanalyzed the localization of MPK4:GFP in the female gametophyte and surrounding ovule tissue by confocal microscopy, and could significantly increase the resolution and image quality (Fig. 2K-P). As a matter of fact, we found MPK4 localizes to the synergid nucleus and to the cytosol (Fig. 2P). Following the results of a nuclear and cytosolic MPK4 localization in the female gametophyte, we are not keeping up with the claim of a translocation of MPK4 from the nucleus to the cytosol in the synergids after PT-reception.

5. The ROS staining results at different FG stages should be provided, which will help the readers see clearly that the abnormal level of ROS starts from FG7.

In **S-Fig.2A-B, E-F**, we analyzed the ROS accumulation in WT, *summ2*, *mpk4* and *mpk4summ2* at the FG4-FG5 stage by carrying out NBT and DAB staining. We found an increased ROS level in *mpk4* and at a lower extent in the *mpk4 summ2*, specifically at the micropylar region after NBT staining, and in the entire ovule and FG after DAB staining. This result shows that the ROS level in *mpk4* is elevated throughout the ovule formation and female gametogenesis.

6. Fig 4 shows that pollen tube rupture is compromised in *mpk4 summ2*, but there is no direct evidence that the phenotype is related to ROS. The author should compare the ROS levels in ovules of *mpk4*, WT, and *mpk4 summ2* before and after pollination.

Thank you for making us aware of that aspect. We analyzed the ROS accumulation before pollination in **Fig. 3A-F** and **Fig.4G-I**. After pollination and fertilization, the ROS accumulation resembles the findings in the unfertilized ovules. After NBT-staining we found a moderately increased ROS accumulation in *mpk4/summ2* (**S-Fig.2C-D**). The results after DAB staining are very variable reaching from strong staining to almost unstained after DAB staining (**S-Fig.2G**).

7. The conclusion in Line123 is not accurate. Thank you for that advise. We removed that conclusion because it was also misplaced at that position.

8. Model in Fig 4F needs to be revised. The position of PT rupture is incorrect and it is not right to draw two synergids, endosperm and embryo appear at the same time. The model can be presented separately, according to before and after receiving pollen tube reception.

As we described the subject in a concise way in the result and discussion part and feel like its interpretation into a model can lead to ambiguity and misinterpretation. As such, we do not wish to continue to present the results in the form of a model at this time.

9. Overall, the quality of the pictures in the article needs to be improved (Fig 2O, 2Q, 2S), and the statistical analysis needs to be more rigorous (Fig 2R).

revised

Minor points :

1. Is Fig 1K the result of CSLM or DIC? Please indicate clearly. revised

2. Why does the cell marked by red fluorescence in Fig 1M look obviously larger than that in Fig 1L? Are the ovules of Fig 1L and 1M at the same developmental stage? I can see the sizes of the ovules are different. Please clarify it.

The focal plane and the angle of observation is different owing to the position of the ovule on the object slide and the position of the nuclei in the FG.

3. The NBT-occurrence (%) of EC in *mpk4* is lower compared to WT in Fig 2D, but the results of NBT staining in Fig 2B/2D seemed different from statistical results.

We tried to present a staining result that provides the most representative impression of strong NBT-staining in the micropylar region.

4. Is Fig 2M-2O the result of DPI or CuCl₂ treatment? These images show the results of the DPI-treatment, as indicated now in Fig. 3L-N.

5. In Line161, 2C-2D should be changed to 2B-2D. revised

6. In the Legend of Fig 3J-M, please clarify what the white arrowhead or white asterisk means. revised
7. Some figures are not properly referred in the text, such as Fig 2M-P and Fig 4F. Thank you for making us aware of that issue. revised.

Reviewer #2 (Remarks to the Author):

The manuscript submitted by Völz et al. describes the MPK4 and SUMM2 signalling module in the development of the female gametophyte in *Arabidopsis thaliana*. While the infertility of *mpk4* homozygous plants has been previously described, and the role of MPK4 in pollen development is now better understood, little is known about this protein's activity in developing female gametophytes.

The authors conduct a series of experiments to structurally characterise the female gametophyte development and pollen tube reception in *mpk4* mutants. The experiments unequivocally show the *mpk4(-)* synergid cells undergo premature degeneration that also correlates with increased ROS accumulation. Given the well-established role for MPK4 in suppressing autoimmunity, ROS production and PCD under pathogenically unchallenged conditions, the authors hypothesise a similar role for MPK4 in synergid cells. Next, the authors test whether mutating the NB-LRR SUMM2 in an *mpk4* background can rescue the reproductive defects of *mpk4* plants. While the pollen defects seen in *mpk4* plants were still present in *mpk4 summ2* double mutants, the female gametophyte and reproductive defects are lessened in *mpk4 summ2* ovaries, suggesting a role for SUMM2 as a guard for MPK4 downstream targets in synergid cells, similar to what has been described for this signalling module in defence. Finally, Völz et al.

identify an invasive pollen tube phenotype in *mpk4 summ2* ovules, reminiscent of the *fer* and *lre* phenotypes.

The work presented in this manuscript will be of interest to the plant science community as it clarifies the role of MPK4 in the female gametophyte and evidences the parallels between defence and pollen tube reception mechanisms in *Arabidopsis*.

The manuscript is well written and the figures well presented. The methodology and statistical analyses also seem correct.

Editor: This reviewer nevertheless has a major concern with this study that the authors should experimentally address to substantiate most of their claims: demonstrating the reproductive defects in *mpk4(-/-)* ovaries is due to defective synergid cells. It is highly likely that the *mpk4(-/-)* ovaries may produce high levels of ROS in other maternal tissues that would impair the pollen tube journey towards the ovule. For this reason, the authors should carry out a series of experiments with hemizygous *mpk4(+/-)* ovaries to demonstrate the gametophytic origin of the *mpk4* maternal reproductive defects.

We totally agree on the importance to confirm that the reproductive defect observed in *mpk4/-* is caused by defective synergids. We believe to have convincing evidence that the reproductive defects are due to defective synergids cells.

By performing PT-attraction assays, we have proven that the PT journey within the transmitting tract and maternal tissue is not broadly perturbed in *mpk4/-* (Fig. 1B-C, E-F). However, once arrived at and grown on the funiculus, the PT in *mpk4*, seems to lose guidance right in front of the ovule and cannot reach the female gametophyte in the great majority of analyzed samples. Semi-in vivo PT attraction assays additionally confirmed that the short-track PT attraction is impeded in *mpk4/-* (Fig. 1D, G-H).

Interestingly, the female gametophytic effect in *mpk4*, concerning premature synergid degeneration, is of sporophytic origin following the results of the transmission-efficiency analysis (Fig. 2J) and the study of the heterozygote *mpk4/+* mutant (Fig S3A-F). The reason for the pre-mature degeneration originates from the high ROS accumulation in the sporophytic micropylar region of the ovule right in front of the FG's synergids. The reduction of the high ROS levels by ROS scavengers and by introducing the *summ2* mutation in *mpk4* restores synergid maturation and function.

Our results show that the MPK4-SUMM2 signaling cascade coordinates the ROS homeostasis in the micropylar ovule region, including synergids, thereby enabling correct synergid maturation as a precondition for successful PT attraction.

Major revisions that should be addressed to provide solid evidence to support the author's claims:

Line 81 – High rates of untargeted ovules could also originate from problems in pollen grain hydration and germination at the stigma, a problem in pollen tube growth through carpel tissues, etc. Authors should show whole GUS-stained ovaries. If *mpk4* is exclusively affecting PT attraction from the synergids, the carpels will show fully grown pollen tubes all along the transmitting tract that are not growing into the ovules. If that's not the case, the authors should describe at which particular step along the PT journey the defects arise. If running these experiments reveal that MPK4 is actually controlling pollen tube germination or growth through stigma and style, the authors should then test in semi-in vitro fertilisation experiments whether pollen tubes germinated and grown through WT stigma and styles are attracted by exposed *mpk4*(-/-) ovules.

Thank you very much to raise this important issue of deeper characterization. We performed aniline blue staining and GUS-staining to track whether the PT growth through the stigma and transmitting tract towards the funiculus and the ovules is affected in *mpk4* (-/-). We applied WT pollen and could confirm that the PT growth through the stigma, the transmitting tract and to the funiculus are not broadly perturbed in *mpk4* (Fig. 1B-F, S-Fig1-P-Q). Yet, after the arrival on the funiculus, we hardly found any PTs that actually reached the ovule/FG but in most cases, the PT seemed to lose guidance (Fig. 1C, E-F). Additionally, semi-in vivo PT-attraction analysis confirmed that *mpk4* -/- ovules are affected in PT-attraction (Fig. 1D, G-H). Our results demonstrate that the PT journey is just impeded right in front of the ovule on the funiculus.

Line 113 - Since *mpk4* homozygous mutant displays widespread deleterious phenotypes, before drawing any conclusions, it is key to demonstrate that the synergid cell defects are gametophytic, and rule out a sporophytic effect caused by surrounding maternal tissues. Secondly, the authors need to show the gametophytic effect of *mpk4* mutation. Repeat the pollination and GUS staining experiment with *mpk4*(+/-) ovaries, in which they should see around 50% of ovules with PT reception (*mpk4*(+) female gametophytes), and 50% without (*mpk4*(-) female gametophytes). And finally, repeat the synergid cell integrity analysis in an *mpk4*(+/-) background, either by CLSM of fixed ovules, or simply with one of their synergid-specific fluorescent markers, in which, again, they should register approximately a 50% increase in the ovules displaying clear synergid cell signal 48h post emasculation. If the experiment yields different results, the authors should reinterpret their results and change their conclusions and discussions accordingly. In addition, the authors could complement the *mpk4*(-/-) mutant with a synergid cell-specific promoter construct (LREpro:MPK4, for example) to demonstrate that the reproductive defect stems from the synergids and not from other sporophytic tissues.

To reveal the inheritance pattern of *mpk4*, we performed reciprocal crosses with WT plants (Fig.2J) and determined the female and male transmission efficiency (TE_F). The TE_F was close to 100% for the transmission of *mpk4* through the female and male gametophyte which shows that the *mpk4* male and female gametophytic transmission efficiencies are not affected. This finding indicates a sporophytic origin of the synergid pre-maturation defect in *mpk4* homozygote mutants. The transmission rate of the self-crossed *mpk4* heterozygote mutant basically mirrors a Mendelian segregation of 1:2:1 (S-Fig. 3F). Furthermore, we showed that the *mpk4* heterozygote mutant is not affected concerning the premature synergid degeneration, seed formation and synergid-marker expression (Fig. 1Q, S-Fig. 3A-E). Thus, we feel the complementation of the *mpk4* mutant by a synergid-specific expression of a rescue-construct is not required at that stage of experimental analysis and we would like to refrain from considering that experiment in the frame of that manuscript.

The authors should provide an extra figure in which they show detailed analysis of the expression pattern of pMPK4 in the ovules, at the two stages they have analysed (FG5, FG7) grown and emasculated under their same conditions. A nuclear-localised fluorescent marker should be used for this experiment to provide the

cellular resolution needed within the ovule. Line 217 – Images in Figure 2Q-S are not clear enough. Authors need to provide confocal images from the ovules expressing MPK4-GFP. And authors should not forget to image in parallel a WT control to compare the autofluorescence in the ovular tissues, that is generally pretty obvious and can be misleading when analysing the MPK4-GFP signal, especially if the synergids undergo degradation.

We reanalyzed the localization of MPK4:GFP in the female gametophyte and surrounding ovule tissue by confocal microscopy, and could significantly increase the resolution and image quality (Fig. 2K-P, WT control S-Fig. 1O). For the co-localization study, we used the nuclear staining dye DAPI. We found MPK4 localizes to the synergid, egg cell and CC nucleus and to the cytosol (Fig. 2N). Following the results of a nuclear and cytosolic MPK4 localization in the synergids and partly, egg cell and CC, we are not maintaining our claim of a translocation of MPK4 from the nucleus to the cytosol in the synergids after PT-reception.

Furthermore, we detected strong MPK4-GFP signal in the entire sporophytic ovular tissue surrounding the FG (Fig. 2K-L).

Revisions that should be addressed to improve the quality of the presented results and discussion:

Line 259 - In this experiment (Figure 3F), the control should have included *mpk4* single mutant as analysed in Figure 1. We updated Figure 4F.

Line 380 - How can the authors argue that the MPK4-SUMM2 module is involved in PT reception when the SUMM2 single mutant has a wild-type PT reception phenotype? If the synergid CrRLK1Ls signalled via MPK4-SUMM2 to trigger synergid cell degeneration after PTs arrive, then *summ2* single mutants should display a PT overgrowth phenotype, as well as the *mpk4/summ2* double. Authors should address this issue in the discussion.

Thank you for raising this interesting issue. Indeed, we found that the *mpk4/summ2* double mutant exhibits a PT reception defect. Yet, in the *summ2* single mutant, redundant factors, whose function depends on MPK4, might compensate for the lack of SUMM2 and enable a regular PT-reception. The disruption of MPK4 in *summ2* might compromise additional guard proteins or regulatory factors that eventually interfere with synergid function and PT-reception. In this regard, the knock-out of the TNL immune receptor SMN1/RPS6 [3, 4] and of the DEAD-Box RNA Helicase SMN2/HEN2 were shown to partly restore the *mpk4* mutant [3], reminiscent to *mpk4/summ2*. Mutations in SMN1/RPS6 and SMN2/HEN2 partially suppressed ROS-accumulation and the dwarf, and autoimmune phenotype of *mpk4* plants. Thus, the structurally distinct NLR proteins, SMN1/RPS6, SMN2/HEN2 and SUMM2, monitor the integrity of the MPK4 pathway and might exert distinct impact on PT-perception following the disruption of MPK4. We added the discussion of this issue to the manuscript.

Line 395 - The results presented by the authors do not point at the connection between a pathogenic infection, and MPK4-SUMM2 reducing fertility as a response. Such claims may be misleading as the authors presented no evidence supporting such connection between systemic resistance and fertility decrease via MPK4. What the authors are presenting is rather a story of how signalling cascades can be co-opted in different contexts, to trigger PCD upon pathogens, or to trigger PCD in ovules in synergid cells upon PT reception.

Thank you for giving us the option to clarify this aspect. In this part of the discussion, we do not intend to claim that a tight connection between pathogenic infection and the control about fertilization is mediated by the MPK4-SUMM2 module. We discuss our findings from a broader point of view and we are keen to introduce related processes that might be associated and thus worth studying in up-coming projects. Maybe, we described these thoughts in an inappropriate manner. Therefore, we revised those paragraphs at critical sites.

I would suggest the authors rather discuss the connection between pathogen responses and pollen reception (ROS production, Ca²⁺ signalling, FERONIA and LLG proteins, NORTIA, and now MPK4 and SUMM2) from an

evolutionary standpoint.

Thank you for that suggestion. We included the discussion of this issue to the manuscript

Minor revisions/comments:

Line 37 - This describes *Arabidopsis thaliana* female gametophyte development, but not many other angiosperms or other types of plants. Please rephrase. revised

Line 43 - Sentence needs rewriting - missing verb: revised

Line 55 - What is male-cytokinesis? rephrase to give more detail about the described role of MPK4 in pollen development

revised

Line 56 - In which pathway does this occur? in the reproductive context? In a defence pathway? Be more precise to aid reader understand the background. MPK4 doesn't necessarily have to interact with the same signalling partners in different developmental contexts. Thank you for that advise. We revised this part of the introduction.

Line 93 – Rephrase “joins together” for better understanding. I suggest: “This marker allows simultaneous characterisation of synergid cell, egg cell, and CC in a single construct”. Thank you for this advice. We replaced our sentence.

Line 116 - Why is this panel in Figure 2 that focuses on ROS production? would be better fitting as part of the general description of the phenotype of *mpk4* in Figure 1. Or split Figure 1 into two so that it does not contain so many panels.

We moved the silique analysis to Figure 1 (Fig. 1Q).

Line 123 - Authors have to add solid evidence to substantiate this claim. At the moment it is speculation.

In the light of the new findings after the revision, we removed that sentence at that misplaced position.

Line 161 - Wording: intense, instead of intensive. revised

Line 171 - The images provided are not of enough quality/resolution to say the staining signal comes from FA or not. Authors should rephrase and say there is strong signal at the micropylar region.

revised

Line 259 - This does not necessarily suggest anything about the synergids in particular, since it has not been shown that the pollen tubes are reaching the ovules normally in *mpk4* ovaries. SUMM2 could be reducing autoimmunity phenotypes along the carpel tissues, and therefore allowing pollen tubes to germinate and grow through them and be attracted towards the ovules.

By performing the appropriate experiments (Fig. 1B-H), suggested by our reviewers, we could show that the journey of the PT through the transmitting tract towards the ovule is not affected in *mpk4*. Thus, our suggestion that that “SUMM2 rescued the pre-mature synergid-PCD observed in the *mpk4* single mutant” is reasonable.

Line 354 - Authors have not discovered that. Authors have discovered MPK4 and SUMM2 affect synergid cell

integrity and possibly the pollen tube reception process. The authors have not provided any evidence for the additional MAPK cascade elements in the synergids – please rephrase to avoid misleading the reader.

We agree and revised that sentence.

Line 379 - FER is a CrRLK1L. I suggest re-writing into: “It is conceivable that a similar CrRLK1L complex modulates SUMM2-mediated cell death in the synergids in response to PT-reception”. PT reception in synergid cells depends on other CrRLK1Ls: HERK1 and ANJ, that interact with LRE and FER, in a proposed trimeric signalling complex. The authors should comment that such FER, LRE, HERK1/ANJ trimeric complex could be the synergid cell counterpart to the immunity FER, LRG1, LET1, LET2 complex.

revised

Line 407 - Please indicate the T-DNA stock IDs for mpk4 and summ2 lines.

revised

Line 515 – Figure S1 should be split into two Supp figures. First one regarding the genetic reporters, and the second regarding the seed set and pollen size

revised

1. Volz, R., et al., *Ethylene signaling is required for synergid degeneration and the establishment of a pollen tube block*. Dev Cell, 2013. **25**(3): p. 310-6.
2. Sun, Y., et al., *Plant egg cell fate determination depends on its exact position in female gametophyte*. Proc Natl Acad Sci U S A, 2021. **118**(8).
3. Takagi, M., et al., *Arabidopsis SMN2/HEN2, Encoding DEAD-Box RNA Helicase, Governs Proper Expression of the Resistance Gene SMN1/RPS6 and Is Involved in Dwarf, Autoimmune Phenotypes of mekk1 and mpk4 Mutants*. Plant Cell Physiol, 2020. **61**(8): p. 1507-1516.
4. Takagi, M., et al., *Disruption of the MAMP-Induced MEKK1-MKK1/MKK2-MPK4 Pathway Activates the TNL Immune Receptor SMN1/RPS6*. Plant Cell Physiol, 2019. **60**(4): p. 778-787.

REVIEWERS' COMMENTS

Reviewer #1 (Remarks to the Author):

This is a revised manuscript reporting that ROS homeostasis in two biological processes, i.e., premature synergid cell death suppression and pollen tube reception, is mediated by MPK4 and SUMM2. The added data in this revised version consolidated the role of MPK4 and SUMM2 in mediating ROS level during the synergid cell death. However, for its role in pollen tube reception, I still see no direct evidence to connect ROS homeostasis to the pollen tube overgrowth phenotype in the *mpk4/summ2* mutant. In Fig4 G-I, it is shown that the ROS level in *mpk4/summ2* ovules was not decreased but rather moderately increased compared with that in the WT ovules. This is not consistent with the phenotype observed in *fer-4* mutant. Thus it seems that the PT overgrowth phenotype observed in the *mpk4/summ2* could not be explained by defects in ROS homeostasis. I think the authors should clarify the link between MPK4/SUMM2-mediated ROS homeostasis and pollen tube reception.

Minor point:

Line 33, 'plant germ cells are formed in...female gametophytes.' Please rephrase.

Reviewer #2 (Remarks to the Author):

Attached a pdf formatted version of the comments. Please download.

Dear Völz and colleagues,

The revised version of the manuscript has been greatly improved, including a series of experiments that strengthen the initially-presented results and allow better informed hypothesis about the roles of MPK4 and SUMM2 during ovule development and reproduction in *Arabidopsis*. This reviewer would like to firstly thank the authors for adding the suggested data, and congratulate them on the insightful piece of work they have constructed by meticulously studying the ovule of *Arabidopsis* which always requires incredible amounts of painstaking work. Please, find below my general comments, interpretation of results and interest, and some additional suggestions to further polish this manuscript. Most of the suggestions below I leave up to the authors and editorial team to be implemented or not in the manuscript prior to publication.

General Comments

The revised version of the manuscript by Völz et al. has greatly improved the microscopy data presented in the original version and replaced it with detailed confocal work. Besides, the authors have presented semi-in vivo pollen tube growth assays as well as full-ovary pollen tube tracking in aniline blue staining that clearly show that WT pollen tubes grow normally through maternal tissues but lack short-range attraction towards the micropyle entrance.

Importantly, the detailed transmission efficiency analysis in *mpk4* (-/-) and *mpk4* (+/-) has clearly indicated that the female reproductive defect of *mpk4* plants has a sporophytic origin. This novel finding profoundly impacts the manuscript's interpretation of results, which originally hinted a synergid-specific, gametophytic MPK4-SUMM2 pathway controlling both FG development and pollen tube reception. In the light of the new evidence, a new scenario emerges.

In this reviewer's opinion, the community will find it very interesting to know that there is a novel pathway of sporophytic maternal control of fertility that employs MPK4 and the guard SUMM2. Speculatively, it is possible that plants can employ this mechanism to negatively regulate reproduction upon challenging conditions (of pathogenic nature, or otherwise). The pollen tube overgrowth detected in *mpk4 summ2* double mutants in a percentage of the ovules is as of today less characterised. Given that the pollen overgrowth only happens in a percentage of ovules, and the *mpk4 summ2* double mutant already show a variety of ovular/seed fates (some aborted, some unfertilised, some untargeted, some properly fertilised, etc), it is very hard to draw any conclusions about the role of

MPK4 or SUMM2 in this particular regard without a much more detailed characterisation of this phenotype. While the authors have done an extensive listing of possible speculative scenarios explaining this, this reviewer's interpretation would be the following:

SUMM2 as well as other NBS-LRRs guard the correct function of a phosphorylation cascade including MEKKs and MPK4 so that when pathogens strive to impair its functionality, SUMM2 will trigger localised cell death and hinder infection progression. However, it is important to differentiate the multiple roles of the kinase MPK4 in the cell from the SUMM2-triggered cell death. In *mpk4* mutant plants, two types of defects appear, those caused by the direct lack of MPK4 in the cell, and those as a consequence of SUMM2/NBS-LRRs triggering cell death because the MPK4 pathway has been disturbed. As such, when a defect present in *mpk4* plants can be rescued by *summ2*, it means that such defect was most likely caused by the SUMM2 surveillance. Therefore, only *mpk4* defects that cannot be complemented by mutation of the NBS-LRRs hold potential for unravelling MPK4-specific functions in the plant. In the framework of the present manuscript, the pollen tube overgrowth falls into the latter category, a reproductive defect that is not caused by SUMM2 (as the *summ2* mutant reproduces normally), but is present in the *mpk4 summ2* double mutant. I would hypothesise that, since MPK4 is expressed both in synergids and integuments, and the SUMM2-induced reproductive impairment is sporophytic, it is possible that MPK4 is key for FER/LRE/NTA signalling within the synergid, and is guarded by SUMM2 in the integuments as a safekeeping mechanism. That way, only when *summ2* is mutated, the FG function for *mpk4* is revealed as pollen tubes overgrow in *mpk4 summ2* ovules. Alternatively, as the authors mention in the discussion, maybe MPK4 is controlling pollen tube discharge from the integuments similarly to what has been reported for JAGGER. However, given the cytosolic localisation of MPK4 (versus the extracellular, and therefore in direct contact with pollen tube and synergids of JAGGER), I find the latter a less plausible explanation for the *mpk4* pollen tube overgrowth defect.

If the above interpretation sounds logical to the authors, it could be incorporated within the discussion. If not, that is also OK.

Experiments to Strengthen the Manuscript

If the authors have the data available or if the other reviewers or handling editors consider these requests crucial, please provide the following. Otherwise, this reviewer will not delay the publication of this story if providing these data requires months

Line 312-317 - Why wasn't the reciprocal cross performed bidirectionally on Figure 1? It is missing the cross in the other direction, pollinating WT with *mpk4* pollen. That way the fact that *mpk4* pollen is also totally impaired would not be a matter of speculation and would allow understanding if SUMM2 complements partially the FG defect but does not restore the pollen defect of *mpk4*.

Lines 360-372 - The article would benefit greatly from SUMM2-GFP or a promoter reporter line of SUMM2. It would be extremely helpful to know if SUMM2 is expressed in integuments only, in FG only, in all ovular tissues like MPK4.

Comments on Figures

Please correct/improve the following

In general, for all figures – In the light of the new evidence, the *mpk4* reproductive defect is sporophytic, most likely originating from the integument layers of the ovule that are most ROS-stained at the micropyle. Ideally, all ovule-containing figures presented should not have been cropped to show the FG only, and should also include the micropyle and integuments. Since I assume the figures were cropped to the FG during manuscript preparation, it may be possible to show this without any repetition of experiments.

Figure 1; Panels B, E, F - micrographs in Figure 1 panels B, E, F are not ideal for publication. Pollen tubes stained with aniline blue should be much more easily distinguishable from the background staining of the rest of ovary structures.

Figure 2; Panels K-L – Which stage FG is this?

Supplementary Fig 1; Panel O - In Figure supp 1O, the confocal image of the WT ovule with background fluorescence control needs to be flipped and rotated, to match the respective WT light microscopy image.

Figure 4 – (And the same goes for other figures in the manuscript) Figure organisation is slightly chaotic and hinders the understanding of the results. In Figure 4, for instance, the panels are

organised following no consistent direction. I assume this is due to a strict number of figures allowed by the journal/type of article. I strongly suggest the authors discuss with the editors making an exception and allowing one or two more main figures. Otherwise please find a more logical panel disposition within figures.

Comments on Wording and Interpretation of Results

Please find this reviewer's suggestions to improve the manuscript understandability and avoid overinterpretation of results

Line 129 – Maybe replace PCD for “undergo premature degeneration”? Isn't PCD a specific term, with different categories of genetic and biochemical signatures? Evidence provided by this piece of research comprises observing more dense cellular structures on the confocal and lack of nuclei. I am not sure this substantiates PCD, and therefore recommend the authors to use “cell degeneration” instead.

Line 130 – Results presented do not allow concluding that it enables each of those processes. What can be concluded from the manuscript's results is that it enables PT entrance to the FG, most likely by maintaining FG integrity at later stages of development and therefore allows the secretion of attractants. Whether MPK4 enables processes beyond that point cannot be affirmed as pollen tubes never make it past the short-range attraction step.

Line 192 - For manuscript consistency purposes: At this particular stage of the narrative, the authors have not determined to which sporophytic tissues you can assign MPK4 activity. Would be better to drop the sentence at “sporophytic origin”. And add that to understand where exactly MPK4 is acting, the authors observed MPK4-GFP in the ovule (explained in the next section).

Line 244-252 - How can the authors say where specifically the ROS staining is originating from in mpk4 mutants when they themselves have made a point that in mpk4 mutants synergids degenerate and are hard to distinguish in the CLSM confocal methodology that is designed to give the maximum resolution of that set of cells? It would be best if the authors mentioned what I assume can be inferred from the microscopy images of mpk4 ROS-stained ovules: micropyle region, CC region, antipodal region...

Lines 355-356 - Misleading wording. The results do not indicate an immune-related defence response. It is a similar cellular response (ROS production in high quantities that is toxic for the cells) using the SUMM2 and MPK4 pathway. It has not been tested under pathogenically-challenged conditions. All that could be said is that the same module is used in pathogenesis and possibly during ovule development.

Line 358 – I suggest exchanging PCD for cell degeneration

Line 410 – Wording: “linked to plant immunity” I suggest to remove it, does not make sense in the syntax of the sentence nor does it resonate with the content of this article.

Line 412 – Wording: The fact that FER regulates pollen tube reception doesn't imply profound mechanisms in reproduction, hormone signalling and immunity...

Line 428 – Wording: Which is the interplay between MPK4 and FER/LRE? with the current wording it is implied that there is an already established interplay between these two signalling elements. Sentence to be rewritten to clarify it is author's speculation.

Line 433 – Wording: “as part of the SUMM2-mediated immunity response”. Again, in my opinion, the fact that the protein SUMM2 has a physiological function in a plant cell, doesn't automatically mean that the response has something to do with immunity. The process in which a function is first assigned to a particular gene/protein does not preclude that same gene/protein from performing a similar cellular function in a different context. If this make sense to the authors, I suggest the “mediated immunity response” is removed. For instance, FER was first identified as a regulator of pollen tube perception in ovules. Since then, FER has been shown to be involved in many other processes and in different organs, immunity in leaves, cell expansion and cell wall integrity in roots. Wouldn't the authors find it strange if they came across a manuscript saying something like: “we have now characterised that the cell expansion in roots is a FER-mediated fertility response” ... ?

Line 441 – Wording: “which and couples” Not sure what the authors meant here; Which uncouples, maybe?

Lines 445-446 - Wording: I suggest it to be rephrased as “... that FER and the closely related LET1 and LET2 form a specific...”

Lines 500-501 - Wording: Repeated “on the other hand” twice on a row.

Sincerely,
Sergio Galindo-Trigo, PhD
University of Oslo

REVIEWERS' COMMENTS

Reviewer #1 (Remarks to the Author):

This is a revised manuscript reporting that ROS homeostasis in two biological processes, i.e., premature synergid cell death suppression and pollen tube reception, is mediated by MPK4 and SUMM2. The added data in this revised version consolidated the role of MPK4 and SUMM2 in mediating ROS level during the synergid cell death. However, for its role in pollen tube reception, I still see no direct evidence to connect ROS homeostasis to the pollen tube overgrowth phenotype in the *mpk4/summ2* mutant. In Fig4 G-I, it is shown that the ROS level in *mpk4/summ2* ovules was not decreased but rather moderately increased compared with that in the WT ovules. This is not consistent with the phenotype observed in *fer-4* mutant. Thus it seems that the PT overgrowth phenotype observed in the *mpk4/summ2* could not be explained by defects in ROS homeostasis. I think the authors should clarify the link between MPK4/SUMM2-mediated ROS homeostasis and pollen tube reception.

We revised title, abstract and discussion regarding that issue following the suggestion of the editor.

Minor point:

Line 33, 'plant germ cells are formed in...female gametophytes.' Please rephrase.

Thank you for giving us this advice. We revised that sentence to introduce the general topic in a more suitable and precise way.

Reviewer #2 (Remarks to the Author):

Attached a pdf formatted version of the comments. Please download.

Dear Völz and colleagues,

The revised version of the manuscript has been greatly improved, including a series of experiments that strengthen the initially-presented results and allow better informed hypothesis about the roles of MPK4 and SUMM2 during ovule development and reproduction in *Arabidopsis*. This reviewer would like to firstly thank the authors for adding the suggested data, and congratulate them on the insightful piece of work they have constructed by meticulously studying the ovule of *Arabidopsis* which always requires incredible amounts of painstaking work.

Please, find below my general comments, interpretation of results and interest, and some additional suggestions to further polish this manuscript. Most of the suggestions below I leave up to the authors and editorial team to be implemented or not in the manuscript prior to publication.

General Comments

The revised version of the manuscript by Völz et al. has greatly improved the microscopy data presented in the original version and replaced it with detailed confocal work. Besides, the authors have presented semi-in vivo pollen tube growth assays as well as full-ovary pollen tube tracking in aniline blue staining that clearly show that WT pollen tubes grow normally through maternal tissues but lack short-range attraction towards the micropyle entrance.

Importantly, the detailed transmission efficiency analysis in *mpk4* (-/-) and *mpk4* (+/-) has clearly indicated that the female reproductive defect of *mpk4* plants has a sporophytic origin. This novel finding profoundly impacts the manuscript's interpretation of results, which originally hinted a synergid-specific, gametophytic MPK4-SUMM2 pathway controlling both FG development and pollen tube reception. In the light of the new evidence, a new scenario emerges.

In this reviewer's opinion, the community will find it very interesting to know that there is a novel pathway of sporophytic maternal control of fertility that employs MPK4 and the guard SUMM2. Speculatively, it is possible that plants can employ this mechanism to negatively regulate reproduction

upon challenging conditions (of pathogenic nature, or otherwise). The pollen tube overgrowth detected in *mpk4 summ2* double mutants in a percentage of the ovules is as of today less characterised. Given that the pollen overgrowth only happens in a percentage of ovules, and the *mpk4 summ2* double mutant already show a variety of ovular/seed fates (some aborted, some unfertilised, some untargeted, some properly fertilised, etc), it is very hard to draw any conclusions about the role of MPK4 or SUMM2 in this particular regard without a much more detailed characterisation of this phenotype. While the authors have done an extensive listing of possible speculative scenarios explaining this,

this reviewer's interpretation would be the following:

SUMM2 as well as other NBS-LRRs guard the correct function of a phosphorylation cascade including MEKKs and MPK4 so that when pathogens strive to impair its functionality, SUMM2 will trigger localised cell death and hinder infection progression. However, it is important to differentiate the multiple roles of the kinase MPK4 in the cell from the SUMM2-triggered cell death. In *mpk4* mutant plants, two types of defects appear, those caused by the direct lack of MPK4 in the cell, and those consequence of SUMM2/NBS-LRRs triggering cell death because the MPK4 pathway has been disturbed. As such, when a defect present in *mpk4* plants can be rescued by *summ2*, it means that such defect was most likely caused by the SUMM2 surveillance. Therefore, only *mpk4* defects that cannot be complemented by mutation of the NBS-LRRs hold potential for unravelling MPK4-specific functions in the plant. In the framework of the present manuscript, the pollen tube overgrowth falls into the latter category, a reproductive defect that is not caused by SUMM2 (as the *summ2* mutant reproduces normally), but is present in the *mpk4 summ2* double mutant. I would hypothesise that, since MPK4 is expressed both in synergids and integuments, and the SUMM2-induced reproductive impairment is sporophytic, it is possible that MPK4 is key for FER/LRE/NTA signalling within the synergid, and is guarded by SUMM2 in the integuments as a safekeeping mechanism. That way, only when *summ2* is mutated, the FG function for *mpk4* is revealed as pollen tubes overgrow in *mpk4 summ2* ovules. Alternatively, as the authors mention in the discussion, maybe MPK4 is controlling pollen tube discharge from the integuments similarly to what has been reported for JAGGER. However, given the cytosolic localisation of MPK4 (versus the extracellular, and therefore in direct contact with pollen tube and synergids of JAGGER), I find the latter a less plausible explanation for the *mpk4* pollen tube overgrowth defect.

If the above interpretation sounds logical to the authors, it could be incorporated within the discussion. If not, that is also OK.

Thank you for the insightful feedback. We agree in the merit of this interpretation and have included portions of it within the discussion to supplement presented scenarios.

Experiments to Strengthen the Manuscript

If the authors have the data available or if the other reviewers or handling editors consider these requests crucial, please provide the following. Otherwise, this reviewer will not delay the publication of this story if providing these data requires months

Line 312-317 - Why wasn't the reciprocal cross performed bidirectionally on Figure 1? It is missing the cross in the other direction, pollinating WT with *mpk4* pollen. That way the fact that *mpk4* pollen is also totally impaired would not be a matter of speculation and would allow understanding if SUMM2 complements partially the FG defect but does not restore the pollen defect of *mpk4*.

Previously, it was shown by Zeng et al, 2011, that WT flowers pollinated by *mpk4* pollen remain basically unfertilized and sterile.

Lines 360-372 - The article would benefit greatly from SUMM2-GFP or a promoter reporter line of SUMM2. It would be extremely helpful to know if SUMM2 is expressed in integuments only, in FG only, in all ovular tissues like MPK4.

We found that the knock-out of *summ2* in *mpk4* mainly rescues the premature synergid degeneration defect in *mpk4* which is of sporophytic origin. By performing single-cell RNA-seq analysis of female

gametophyte cells, Song et al found that *SUMM2* seems to not be expressed in female gametophytic cells owed to the lack of detected transcripts. Thus, we assume that *SUMM2* expression is restricted to the sporophytic tissue.

Comments on Figures

Please correct/improve the following

In general, for all figures – In the light of the new evidence, the *mpk4* reproductive defect is sporophytic, most likely originating from the integument layers of the ovule that are most ROS-stained at the micropyle. **Ideally, all ovule-containing figures presented should not have been cropped to show the FG only**, and should also include the micropyle and integuments. Since I assume the figures were cropped to the FG during manuscript preparation, it may be possible to show this without any repetition of experiments.

Thank you for that advise. We agree and present an extended view of the ovule, where it wasn't properly provided. (Fig. 3 d, e, q, r, Fig.4 g, h)

Figure 1; Panels B, E, F - micrographs in Figure 1 panels B, E, F are not ideal for publication. Pollen tubes stained with aniline blue should be much more easily distinguishable from the background staining of the rest of ovary structures.

Thank you for making us aware of that issue. The image settings were not optimal, so we adjusted the image properties thereby providing a better view of the sample.

Figure 2; Panels K-L – Which stage FG is this?

Supplementary Fig 1; Panel O - In Figure supp 1O, the confocal image of the WT ovule with background fluorescence control needs to be flipped and rotated, to match the respective WT light microscopy image.

Panels K-L of Figure 2 show a FG at the stage FG6/FG7. We revised the former Supplementary Fig. 1o, now, it is Supplementary Fig. 1l.

Figure 4 – (And the same goes for other figures in the manuscript) Figure organisation is slightly chaotic and hinders the understanding of the results. In Figure 4, for instance, the panels are organised following no consistent direction. I assume this is due to a strict number of figures allowed by the journal/type of article. I strongly suggest the authors discuss with the editors making an exception and allowing one or two more main figures. Otherwise please find a more logical panel disposition within figures.

We revised Figure 4 and hope having found a more concise figure arrangement in this way.

Comments on Wording and Interpretation of Results

Please find this reviewer's suggestions to improve the manuscript understandability and avoid overinterpretation of results

Line 129 – Maybe replace PCD for “undergo premature degeneration”? Isn't PCD a specific term, with different categories of genetic and biochemical signatures? Evidence provided by this piece of research comprises observing more dense cellular structures on the confocal and lack of nuclei. I am not sure this substantiates PCD, and therefore recommend the authors to use “cell degeneration” instead.

revised

Line 130 – Results presented do not allow concluding that it enables each of those processes. What can be concluded from the manuscript's results is that it enables PT entrance to the FG, most likely by maintaining FG integrity at later stages of development and therefore allows the secretion of

attractants. Whether MPK4 enables processes beyond that point cannot be affirmed as pollen tubes never make it past the short-range attraction step.

Thank you for this advice. Owing to the fact that synergids are necessary for PT attraction, we just laid out possible consequences that are supposed to be taken into account, e.g. PT-attraction, sperm release and gamete fertilization. We rephrased the sentence to clarify this issue.

Line 192 - For manuscript consistency purposes: At this particular stage of the narrative, the authors have not determined to which sporophytic tissues you can assign MPK4 activity. Would be better to drop the sentence at "sporophytic origin". And add that to understand where exactly MPK4 is acting, the authors observed MPK4-GFP in the ovule (explained in the next section).

We revised the sentence.

Line 244-252 - How can the authors say where specifically the ROS staining is originating from in mpk4 mutants when they themselves have made a point that in mpk4 mutants synergids degenerate and are hard to distinguish in the CLSM confocal methodology that is designed to give the maximum resolution of that set of cells? It would be best if the authors mentioned what I assume can be inferred from the microscopy images of mpk4 ROS-stained ovules: micropyle region, CC region, antipodal region...

We revised this paragraph following the reviewer's suggestions.

Lines 355-356 - Misleading wording. The results do not indicate an immune-related defence response. It is a similar cellular response (ROS production in high quantities that is toxic for the cells) using the SUMM2 and MPK4 pathway. It has not been tested under pathogenically-challenged conditions. All that could be said is that the same module is used in pathogenesis and possibly during ovule development.

We replaced "indicate" by "suggest".

Line 358 – I suggest exchanging PCD for cell degeneration

revised

Line 410 – Wording: "linked to plant immunity" I suggest to remove it, does not make sense in the syntax of the sentence nor does it resonate with the content of this article.

revised

Line 412 – Wording: The fact that FER regulates pollen tube reception doesn't imply profound mechanisms in reproduction, hormone signalling and immunity...

revised

Line 428 – Wording: Which is the interplay between MPK4 and FER/LRE? with the current wording it is implied that there is an already established interplay between these two signalling elements. Sentence to be rewritten to clarify it is author's speculation.

Thank you for making us aware of this inaccuracy. We revised this sentence.

Line 433 – Wording: "as part of the SUMM2-mediated immunity response". Again, in my opinion, the fact that the protein SUMM2 has a physiological function in a plant cell, doesn't automatically mean that the response has something to do with immunity. The process in which a function is first assigned to a particular gene/protein does not preclude that same gene/protein from performing a similar cellular function in a different context. If this make sense to the authors, I suggest the "mediated

immunity response” is removed. For instance, FER was first identified as a regulator of pollen tube perception in ovules. Since then, FER has been shown to be involved in many other processes and in different organs, immunity in leaves, cell expansion and cell wall integrity in roots. Wouldn't the authors find it strange if they came across a manuscript saying something like: “we have now characterised that the cell expansion in roots is a FER-mediated fertility response”
... ?

revised

Line 441 – Wording: “which and couples” Not sure what the authors meant here; Which uncouples, maybe?

revised

Lines 445-446 - Wording: I suggest it to be rephrased as “... that FER and the closely related LET1 and LET2 form a specific...”

revised

Lines 500-501 - Wording: Repeated “on the other hand” twice on a row.

revised

=====

Many thanks for the hard work performed by both of the reviews and their insightful contributions to the editing process of the manuscript,

Gratefully yours,
Ronny Völz